# Mamba State-Space Models Are Lyapunov-Stable Learners

**John T. Halloran**                                                    *halloranjt@leidos.com*
*Leidos*

**Manbir Gulati**
*Leidos*

**Paul Roysdon**
*Leidos*

**Reviewed on OpenReview:** *https://openreview.net/forum?id=wzsYQYs3dO*

## Abstract

Mamba state-space models (SSMs) have recently outperformed state-of-the-art (SOTA) Transformer large language models (LLMs) in various tasks and been widely adapted. However, a major concern for stable learning in recurrent-based deep models (such as SSMs) is the sensitivy of their recurrent dynamics. Despite widespread adaptation, the sensitivity of Mamba's recurrent dynamics under common fine-tuning methods–e.g., mixed-precision fine-tuning (MPFT) and parameter-efficient fine-tuning (PEFT)–remains unexplored. Empirically, we show that Mamba LLMs are extremely stable to changes introduced by combinations of MPFT and PEFT, in stark contrast to Transformer LLMs, which we demonstrate may drastically diverge from their respective full-precision counterparts under different combinations of MPFT and PEFT (despite the near-ubiquitous adaptation of these fine-tuning frameworks for attention-based models). The demonstrated robustness of Mamba LLMs are due to their recurrent dynamics, which we prove are guaranteed to be stable using dynamical systems theory (in particular, Lyapunov stability). We conclude by using MPFT and PEFT to novelly study Mamba LLMs' in-context learning (ICL) abilities on *natural language tasks*, thus supplementing other recent work.

## 1 Introduction

Mamba(Gu & Dao, 2024a) state-space models (SSMs) have been shown to greatly outperform comparable attention-based LLMs (Biderman et al., 2023) across a large number of standard natural language benchmarks. Subsequently, pretrained Mamba models have been widely adapted across a large number of data modalities (Liu et al., 2024; Li & Chen, 2024; Quan & Li, 2024; Li et al., 2024), tasks (Xie et al., 2024; Wang et al., 2024a), and architectures (Anthony et al., 2024; Park et al., 2024; Lieber et al., 2024). However, despite such widespread adaptation and subsequent research threads (Dao & Gu, 2024; Park et al., 2024; Wang et al., 2024b), no existing work has sought to understand the stability of learning Mamba SSMs in common LLM fine-tuning frameworks, such as mixed-precision fine-tuning (MPFT). For recurrent-based deep models, such as Mamba SSMs, the sensitivity of recurrent dynamics to small changes (e.g., those introduced by mixed-precision) is a primary concern for stable learning (Bengio et al., 1994; Pascanu et al., 2013).

To determine the stability of time-varying Mamba models, we leverage theory from dynamical systems. Under the framework of Lyapunov stability, we show that small input changes within the SSM layer of both Mamba and the more recent Mamba-2 (Dao & Gu, 2024) models do not lead to exponentially deviating outputs. Furthermore, we extend these theoretical results to account for deviations in input and latent Mamba states produced by combinations of MPFT and PEFT.

We empirically validate these theoretical results; for a large number of randomly generated SSM layers, we show that manually adjusting initial latent and input states produces maximum deviations in the output states which exponentially decrease over discrete time. Furthermore, by expanding previous divergence performance metrics (Dettmers et al., 2022; Dettmers & Zettlemoyer, 2023; Dettmers et al., 2024) and evaluating combinations of MPFT and PEFT, we show that fine-tuned Mamba LLMs do not substantially deviate in performance compared to full-precision full fine-tuning. This is demonstrated in stark contrast to comparable Transformer-based LLMs, which we show may drastically differ from their full-precision full fine-tuning counterparts, exhibiting *large deviation spikes*. **Over two widely used fine-tuning datasets and two widely used natural language benchmarks, attention-based LLMs Pythia** (Biderman et al., 2023) **and OpenELM** (Mehta et al., 2024) **exhibit 9 and 4 large deviation spikes, respectively, whereas Mamba LLMs exhibit zero**. Thus, Mamba LLMs are significantly more stable to performance deviations under MPFT and PEFT, whereas Transformer LLMs are susceptible to large deviation spikes.

Lastly, we use stable MPFT and PEFT to explore the ICL capabilities of instruction tuned Mamba and Mamba-2 models on *natural language tasks*, thus complementing recent studies which evaluated Mamba ICL on synthetic tasks (Park et al., 2024; Lee et al., 2024). While the ICL capabilities of pretrained Mamba models lag behind those of comparable Transformer models–Mamba and Mamba-2 models only achieve 38% and 82%, respectively, of the performance improvements (relative to zero-shot) of Pythia models–instruction tuned Mamba and Mamba-2 models improve to 81.5% and 132% of the ICL improvements achieved by Pythia. Thus, while pretrained Mamba models' ICL capabilities lag behind comparable Transformers, we show that instruction-tuning using MPFT and PEFT greatly narrows this gap.

Our major contributions are summarized as follows:

- We derive bounds on the Lyapunov exponents of both Mamba and Mamba-2 models' SSM equations. Using these bounds, we theoretically show that small input changes within the `MambaBlock` do not lead to exponentially deviating outputs.

- Building on the aforementioned bounds, we theoretically show that changes to `MambaBlock` latent and input states stemming from MPFT and PEFT do not lead to exponentially deviating outputs.

- We empirically demonstrate the above theoretical results:
  - Directly introducing $\varepsilon$ changes to the inputs states of 100 random `MambaBlock`, we confirm that the resulting output states do not deviate from the unperturbed outputs over discrete time (the maximum deviation exponentially decreases).
  - For MPFT and PEFT, we measure the performance deviation between `FP32` full fine-tuned and MPFT/LoRA Mamba models and compare to attention-based models. Across two fine-tuning datasets, two widely used natural language benchmarks, several model sizes, and a large number of MPFT/PEFT configurations, we show that training Mamba LLMs is significantly more stable than comparable Transformer LLMs.

- We expand performance divergence measures previously used to study MPFT and PEFT for Transformer LLMs (Dettmers et al., 2022; Dettmers & Zettlemoyer, 2023; Dettmers et al., 2024). In addition to showing Mamba models are drastically more stable than comparable attention-based LLMs, the expanded divergence metric also demonstrates that Transformer LLMs are susceptible to large deviation spikes.

- Using stable MPFT and PEFT, we complement recent studies by examining the ICL capabilities of Mamba/Mamba-2 models evaluated on *natural language tasks*. We show that instruction tuning allows SSMs to perform ICL competitively with comparable Transformer LLMs on natural language tasks.

**Terminology**. When describing a particular foundation model or result, we use the term "Mamba model" to refer to one of the original models released in Gu & Dao (2024a) and "Mamba-2 model" to refer to models released in Dao & Gu (2024). Despite subtle architectural differences, these two SSMs share the same state-space equations and design scheme of storing the majority of SSM parameters in a large memory buffer.

We thus synonymously use the term `MambaBlock` to refer to the SSM layer of both Mamba and Mamba-2 models.

## 2 Background

**Mixed-precision and parameter-efficient fine-tuning**. In the current era of extremely large foundation models, both MPFT and PEFT have become ubiquitous tools for the rapid adaptation of Transformer-based LLMs towards specific applications. PEFT using adapters (He et al., 2021) allows a large pretrained model to be efficiently adapted for a particular downstream task by freezing the full model and training only a small number of extra parameters. Arguably the most widely used such PEFT method is LoRA (Hu et al., 2021), which injects trainable low-rank matrices into Transformer layers to approximate weight updates.

To further decrease the computational demands necessary for LLM fine-tuning and inference, MPFT via mixed-precision (i.e., `FP16` or `BF16`) (Kalamkar et al., 2019; Micikevicius et al., 2018) and quantized low-precision (Dettmers et al., 2024) have proven effective strategies to reduce GPU memory and runtime requirements without deleterious effects on downstream performance (Dettmers et al., 2024; Wu et al., 2020). Additionally, mixed-precision approaches have paved the way for hardware-aware optimizations within the self-attention module (Dao et al., 2022), greatly mitigating the quadratic complexity of Transformer LLMs. Together, PEFT and MPFT have created a rich ecosystem with which varying combinations of these approaches may be used to meet the computational constraints of a given training system.

**State-space Models**. *Structured state-space sequence* (S4) models (Gu et al., 2022; Fu et al., 2023) are SSMs which leverage linear time-invariant (LTI) systems to combine the computational advantages of Transformers–i.e., highly parallelizable training–and recurrent neural networks (RNNs)–i.e., subquadratic autoregressive inference using recurrency. Within the S4 layer, an input signal is discretized and LTI parameters representing the input's latent dynamics are learned. Owing to the S4 block's latent dynamics being LTI, the S4 block's output may be thus compactly represented as a single convolution between the input and an *SSM convolution kernel* (a matrix whose entries are products of LTI learnable parameters resulting from unrolling the state-space equations). However, despite hardware efficiency and long-dependency-modeling improvements, LTI-based S4 models remained inferior to Transformers of comparable parameter-sizes for natural language tasks, even when augmenting S4 layers with attention-layers for hybrid architectures (Gu & Dao, 2024a).

Innovating on these previous S4 approaches, Mamba utilizes time-*varying* parameters to model latent dynamics, thus broadening the ability to capture nuanced changes evolving in discrete-time. Without LTI dynamics, however, the input-output representation via the SSM convolution kernel is no longer applicable, thus voiding previous hardware-aware S4 optimizations (Fu et al., 2023). To enable hardware efficiency with time-varying SSM parameters, (Gu & Dao, 2024a) thus introduced extensively customized CUDA kernels which implement highly parallelized prefix sums to compute recurrent states. Subsequently, Dao & Gu (2024) considered the unrolled state-space equations and leveraged tensor contractions (i.e., einsum notation (Rogozhnikov, 2022)) to efficiently calculate Mamba variables. The resulting Mamba-2 foundation models contained significantly larger latent-variable dimensions than the Mamba models of (Gu & Dao, 2024a), while maintaining efficiency on modern GPU accelerators.

**In-context learning**. ICL provides an adaptable alternative to fine-tuning. Rather than fine-tune the LLM directly, ICL augments a prompt with $n$ relevant examples (called *shots*) preceding the query of interest. Given sufficiently large models and pretraining data (Brown et al., 2020; Wei et al., 2022), Transformer LLMs have proven adept at learning new concepts on the fly provided such few-shot prompting. However, it is worth noting that ICL inference time increases dramatically as the number of shots grows (due to self-attention's quadratic complexity) and PEFT (when possible) is known to produce more accurate downstream learning results (Brown et al., 2020; Liu et al., 2022).

## 3 Mamba state-space models

For latent-variable dimension $d$ and maximum input sequence length $T$, the `MambaBlock` defines state-space parameters $\mathbf{A}, \mathbf{B}_t, \mathbf{C}_t, \boldsymbol{\Delta}_t \in \mathbb{R}^{d \times d}$ for $t \in \{1, \ldots, T\}$. The matrix $\boldsymbol{\Delta}_t$ controls the discrete step-size. Given an

input sequence $\mathbf{u}_1, \ldots, \mathbf{u}_T \in \mathbb{R}^d$, the following linear mapping through latent states $\boldsymbol{x}_1, \ldots, \boldsymbol{x}_T \in \mathbb{R}^d$ is used to produce the output $\mathbf{y}_1, \ldots, \mathbf{y}_T \in \mathbb{R}^d$:

$$\boldsymbol{x}_t = \bar{\mathbf{A}}_t \boldsymbol{x}_{t-1} + \bar{\mathbf{B}}_t \mathbf{u}_t \tag{1}$$

$$\mathbf{y}_t = \mathbf{C}_t \boldsymbol{x}_t, \tag{2}$$

where $\bar{\boldsymbol{\Delta}}_t = \texttt{softplus}(\texttt{Linear}(\boldsymbol{\Delta}_t)) \in \mathbb{R}^{d \times d}$, $\bar{\mathbf{A}}_t = \exp\left(\bar{\boldsymbol{\Delta}}_t \mathbf{A}\right)$ and $\bar{\mathbf{B}}_t = \mathbf{A}^{-1}(\bar{\mathbf{A}} - \mathbf{I})\mathbf{B}_t$. In practice, $\mathbf{A}, \mathbf{B}_t, \mathbf{C}_t$ and $\boldsymbol{\Delta}_t$ are diagonal matrices.

Due to the time-variance of Equations 1 and 2, previous SSM stability results are no longer applicable (discussed further in Appendix E). The Mamba foundation models were pretrained in full `FP32` precision and, subsequently, official implementations have cautioned against training in reduced precision (Gu & Dao, 2024b; Huggingface, 2024). Thus, how sensitive the `MambaBlock`'s recurrent dynamics are remains an open question. We next answer the latter using theory from dynamical systems.

### 3.1 Stable dynamics in the `MambaBlock`

For Mamba's discrete dynamical system in Equations 1 and 2, define

$$\boldsymbol{x}_t = F_\theta(\boldsymbol{x}_{t-1}, \mathbf{u}_t), \tag{3}$$

where $\theta$ denotes the time-varying parameters described in Section 3. For input sequence $\mathbf{u}_1, \ldots, \mathbf{u}_T$ and initial latent state vector $\boldsymbol{x}_0$, we thus write

$$\boldsymbol{x}_T = F_\theta(F_\theta(\ldots F_\theta(\boldsymbol{x}_0, \mathbf{u}_1))) \coloneqq F_\theta^{T-1}(\boldsymbol{x}_0, \mathbf{u}_1).$$

The rate of divergence between two scalar $\varepsilon$-close inputs to a discrete dynamical system is bounded by the system's maximal Lyapunov exponent $\lambda_{\texttt{max}}$ (Mikhaeil et al., 2022). Given $\lambda_{\texttt{max}}$ and two initial values $(\boldsymbol{x}_0, \mathbf{u}_1)$ and $(\boldsymbol{x}_0 + \varepsilon, \mathbf{u}_1 + \varepsilon)$, the maximum deviation between these points grows as (Laffargue et al., 2013; Sayama, 2015):

$$\max |F_\theta^N(\boldsymbol{x}_0, \mathbf{u}_1) - F_\theta^N(\boldsymbol{x}_0 + \varepsilon, \mathbf{u}_1 + \varepsilon)| \in \mathcal{O}(|\varepsilon| \exp\left(N \lambda_{\texttt{max}}\right)).$$

Thus, when $\lambda_{\texttt{max}} > 0$, nearby trajectories exponentially separate and, when $\lambda_{\texttt{max}} \leqslant 0$, nearby trajectories ultimately converge to the same fixed point or periodic cycles.

The maximal Lyapunov exponent is defined as

$$\lambda_{\texttt{max}} \coloneqq \lim_{T \to \infty} \frac{1}{T} \log \left\| \prod_{t=0}^{T} \frac{\partial \boldsymbol{x}_t}{\partial \boldsymbol{x}_{t-1}} \right\|_2,$$

where $\|\|_2$ denotes the spectral norm for matrices. For an arbitrary `MambaBlock`, we prove the following:

**Theorem 1.** *Let $(\boldsymbol{x}_{t-1}, \mathbf{u}_t)$ be the latent state and input at an arbitrary time $t \in \{1, \ldots, T\}$ within a `MambaBlock`. Then small changes $(\boldsymbol{x}_{t-1} + \varepsilon, \mathbf{u}_t + \varepsilon)$ produce deviations which are exponentially non-increasing over discrete-time. That is, $\max |F_\theta^N(\boldsymbol{x}_{t-1}, \mathbf{u}_t) - F_\theta^N(\boldsymbol{x}_{t-1} + \varepsilon, \mathbf{u}_t + \varepsilon)| \in \mathcal{O}(\varepsilon \exp\left(N\zeta\right))$, for some scalar $\zeta \leqslant 0$.*

The proof of Theorem 1 is available in Appendix A, where the maximal Lyapunov exponent for an arbitrary `MambaBlock` is first proven to be non-positive. The main result subsequently follows.

Thus, the latent states of Mamba and Mamba-2 models are stable under small input changes. However, variables $\mathbf{y}_1, \ldots, \mathbf{y}_T$ are the primary outputs for such models, particularly for LLM applications. We next show that, given Theorem 1, Mamba and Mamba-2 output variables are also stable.

**Theorem 2.** *Assume $(\boldsymbol{x}_{t-1} + \varepsilon, \mathbf{u}_t + \varepsilon)$ produce deviations which are exponentially non-increasing over discrete-time. Then small changes to the output $\mathbf{y}_t$ are also exponentially non-increasing over discrete time.*

The proof of Theorem 2 is available in Appendix B. Thus, by Theorems 1 and 2, the latent and output states of both Mamba and Mamba-2 models are stable to changes encountered during recurrency.

### 3.1.1 Divergence bounds for automatic mixed-precision and PEFT

For an arbitrary `MambaBlock`, let $\boldsymbol{x}_0, \ldots, \boldsymbol{x}_T$ and $\mathbf{u}_1, \ldots, \mathbf{u}_T$ be the trajectory of latent and input states under full-precision fine-tuning. During a forward pass, automatic mixed-precision (AMP) saves time and memory by computing forward activations in half-precision (`FP16` or `BF16`). During a backward pass, AMP computes gradients in half-precision and up-casts to full-precision prior to updating. MFPT thus introduces changes to the inputs $\mathbf{u}_1, \ldots, \mathbf{u}_T$ (which are passed through a `Swish`) and latent states $\boldsymbol{x}_0, \ldots, \boldsymbol{x}_T$ through $\bar{\boldsymbol{\Delta}}_t$ (which is passed through a `softplus`), up/downcasting of parameter updates, and the aforementioned changes to the input states.

Potentially in concert with MPFT, PEFT via LoRA induces changes by learning low-rank matrices $\hat{\mathbf{A}}_t, \hat{\mathbf{B}}_t, \hat{\mathbf{C}}_t$

$$\hat{\boldsymbol{x}}_t = (\bar{\mathbf{A}}_t + \hat{\mathbf{A}}_t)\boldsymbol{x}_{t-1} + (\bar{\mathbf{B}}_t + \hat{\mathbf{B}}_t)\mathbf{u}_t$$
$$\hat{\mathbf{y}}_t = (\mathbf{C}_t + \hat{\mathbf{C}}_t)\boldsymbol{x}_t$$

Let $\boldsymbol{x}'_0, \ldots, \boldsymbol{x}'_T$ and $\mathbf{u}'_1, \ldots, \mathbf{u}'_T$ be the trajectory of latent and input states under MPFT and/or PEFT. We thus have the following two theorems:

**Theorem 3.** *Let $\epsilon_t^u = \max |\mathbf{u}_t - \mathbf{u}'_t|$, i.e., the maximum, positive scalar-difference between the tth MPFT/PEFT and full fine-tuning input state. Similarly, let $\epsilon_t^x = \max |\boldsymbol{x}_t - \boldsymbol{x}'_t|$ and $\epsilon^* = \max \bigcup_{t=0}^{T} \{\epsilon_t^u, \epsilon_t^x\}$ (where we define $\mathbf{u}_0 = 0$). We thus have*

$$\max |F_\theta^T(\boldsymbol{x}_0, \mathbf{u}_1) - F_\theta^T(\boldsymbol{x}'_0, \mathbf{u}'_1)| \in \mathcal{O}(\epsilon^* \exp(T\zeta)),$$

*for some scalar $\zeta \leqslant 0$. Thus, differences introduced by AMP and/or PEFT for Mamba models do not exponentially compound over discrete-time.*

**Theorem 4.** *Let $\mathbf{y}'_t$ be the trajectory of output states produced by $\boldsymbol{x}'_0, \ldots, \boldsymbol{x}'_T$ and $\mathbf{u}'_1, \ldots, \mathbf{u}'_T$ under MPFT and/or PEFT. Then we thus have*

$$\max |\mathbf{y}_T - \mathbf{y}'_T| \in \mathcal{O}(\epsilon^* \exp(T\zeta)),$$

*for some scalar $\zeta \leqslant 0$. Thus, small changes to the output $\mathbf{y}_t$ resulting from MPFT and/or PEFT are exponentially non-increasing over discrete time.*

The proof of Theorem 3 is available in Appendix C. Theorem 4 may be derived by directly applying the proof technique of Theorem 2 to the results of Theorem 3. Thus, the latent and output states of both Mamba and Mamba-2 models are stable to changes encountered during AMP and MFPT.

## 4 Experiments

### 4.1 Non-divergent `MambaBlock` dynamics

We explore the implications of Theorem 2 for arbitrary `MambaBlock`s. For $n$ random trials and various $\varepsilon$, we randomly initialize the inputs (i.e., $u_t$ for $t = 1, \ldots, T$) and parameters (i.e., $\mathbf{A}, \mathbf{B}_t, \mathbf{C}_t, \boldsymbol{\Delta}_t$) of a `MambaBlock` with $T = 2048$ and latent-variable dimension $d = 64$.

For each trial and $\varepsilon$, we first calculate the `MambaBlock` outputs $\mathbf{y}_1, \ldots, \mathbf{y}_T$ (where $\mathbf{y}_t = F_\theta^t(\boldsymbol{x}_0, \mathbf{u}_1)$). We then perturb the initial states $(\boldsymbol{x}_0 + \varepsilon, \mathbf{u}_1 + \varepsilon) = (\boldsymbol{x}'_0, \mathbf{u}'_1)$ and calculate the subsequent perturbed model outputs $\mathbf{y}'_1, \ldots, \mathbf{y}'_T$ (where $\mathbf{y}'_t = F_\theta^t(\boldsymbol{x}'_0, \mathbf{u}'_1) = F_\theta^t(\boldsymbol{x}_0 + \varepsilon, \mathbf{u}'_1 + \varepsilon)$). The maximum deviation is then calculated for each time point, i.e., $\max |y_t - y'_t| = \max |F_\theta^t(\boldsymbol{x}_0, \mathbf{u}_1) - F_\theta^t(\boldsymbol{x}_0 + \varepsilon, \mathbf{u}_1 + \varepsilon)|$. Figure 1 shows both the mean maximum deviation per time averaged across trials, as well as the standard deviation, for $\varepsilon = 0.1$.

As we can see, the maximum divergence starts out proportional to the change in initial conditions, but quickly decays and converges to the original output trajectory. Thus, changes to a `MambaBlock`s inputs do not compound over discrete time. Additional experiments for $\varepsilon \in \{0.01, 0.001\}$ are available in Appendix D and demonstrate the same dynamics.

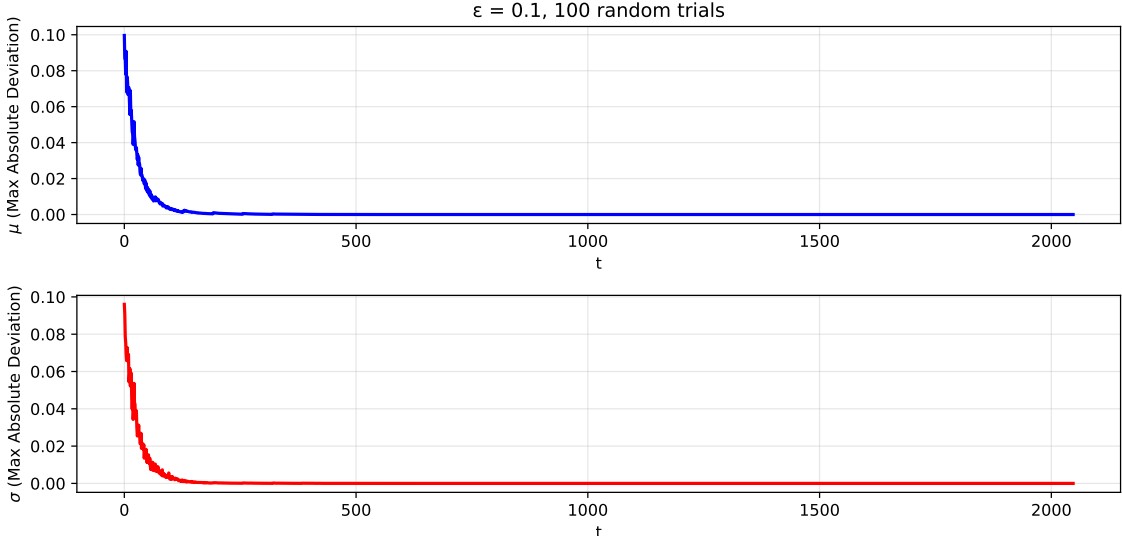

Figure 1: For 100 random `MambaBlock`s, initial latent and input states $(\boldsymbol{x}_0, \mathbf{u}_0)$ are deviated by $\varepsilon$. The resulting maximum deviation in output states is computed and shown to exponentially decrease over discrete time. $\mu$ denotes the maximum absolute deviation at discrete time $t$ averaged over all trials, while $\sigma$ denotes the respective standard deviation at each discrete time.

## 4.2 Non-divergent Mamba MPFT and PEFT

We explore the implications of Theorem 4 for combinations of MPFT and PEFT on pretrained Mamba models. We focus on utilizing LoRA (Hu et al., 2021), which is arguably the most widely used PEFT framework for LLMs. Using the `Alpaca` dataset (Taori et al., 2023), `Mamba 160M`, `410M`, and `790M` models are fine-tuned for three epochs with a maximum sequence length of 512. We denote the targeting of all linear layers (ALL) for LoRA as *ALL LoRA*, the targeting of a subset of linear layers (SLL) for LoRA as *SLL LoRA* (i.e., $\boldsymbol{\Delta}_t, \mathbf{B}_t$, and $\mathbf{C}_t$ parameters), and no adapters as *Full* (i.e., full fine-tuning).

Each fine-tuning run occurred on a single Nvidia A10G GPU (24 GB total memory). To further limit extraneous numerical effects, the same batch size is used for all `FP32`, `FP16`, and `BF16` experiments for a given model size. While this leads to hardware underutilization (i.e., non-saturated GPU memory for mixed-precision and LoRA experiments), this is necessary to guarantee no divergence is due to differences in parameter update schedules. For comparison, two Transformer-based LLM families of similar parameter counts are fine-tuned using the same experimental setup: `Pythia` (sizes `160M`, `410M`, and `1B`) and `OpenELM` (Mehta et al., 2024) (sizes `270M` and `450M`). The training recipe for all models was adapted from Tunstall et al. (2023), with the `AdamW_torch` optimizer and a `cosine annealing` schedule. Further experimental details are available in Appendix F.

All models were evaluated using the LM evaluation harness from Eleuther AI (Gao et al., 2023). **Model performance is measured as percent accuracy** using the MMLU (Hendrycks et al., 2020) and Winogrande (Sakaguchi et al., 2021) datasets. The difference in model performance is reported as the mean *divergence* (i.e., absolute difference) between the original full-precision and respective mixed-precision model, averaged over $\{0, 1, 3, 5\}$-shot percent accuracy. Thus, **a divergence greater than one denotes an average difference greater than one entire percentage of accuracy.** We thus refer to a divergence greater than unity as a *large deviation spike*.

Mean divergence results for all models are displayed in Figure 2(a) and Figure 2(b) for MMLU and Winogrande, respectively. Across mixed-precisions and adapter settings, Mamba displays smaller divergences than both Pythia and OpenELM models. E.g., **for Winogrande evaluations, `Alpaca` fine-tuning with Mamba models is an average 2.9 and 2.4 times smaller in mean divergence than Pythia and OpenELM models, respectively**. For MMLU evaluations, `Alpaca` fine-tuning with Mamba models is an average 2.6

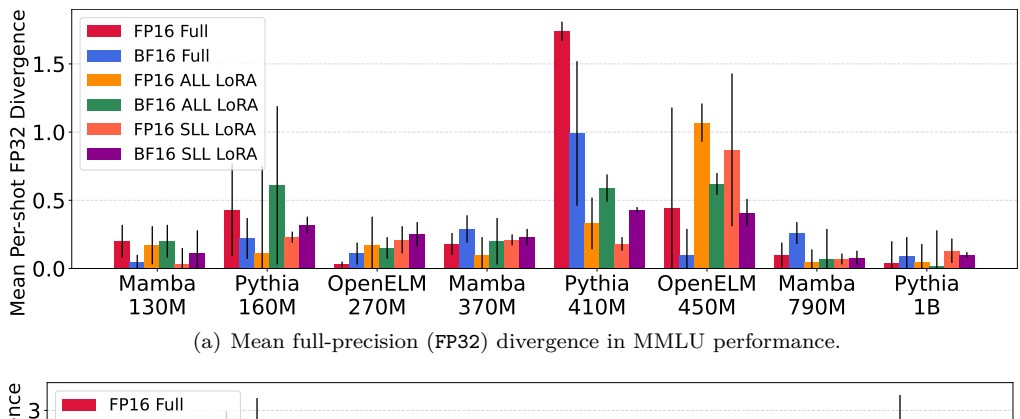

(a) Mean full-precision (`FP32`) divergence in MMLU performance.

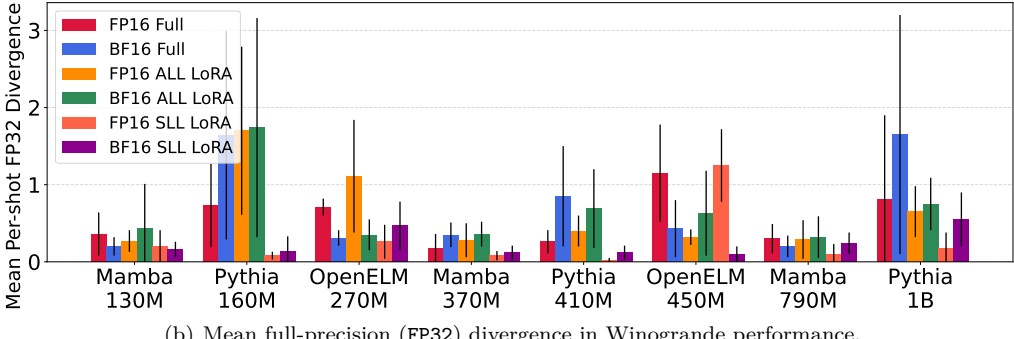

(b) Mean full-precision (`FP32`) divergence in Winogrande performance.

Figure 2: Mean full-precision (`FP32`) divergence in performance for Mamba, Pythia, and OpenELM models. Models are fine-tuned over the `Alpaca` dataset using different combinations of MPFT and PEFT. Full fine-tuning (i.e., no PEFT adapters) is denoted as `Full`.

times smaller in mean divergence than both Pythia and OpenELM models. Importantly, Mamba models do not exhibit large deviation spikes after fine-tuning, in contrast to both Pythia and OpenELM models.

We note that the results in Figure 2 display the fine-tuning of 72 LLMs and 576 few-shot evaluations. In Appendix H, these experiments are expanded to include the `LIMA` fine-tuning dataset (Zhou et al., 2024) as well as the standard deviations for all mean deviation results. Across all experiments, Mamba models are significantly more stable for MPFT/PEFT compared to Transformer-based LLMs. E.g., **for MMLU evaluations, `LIMA` fine-tuning with Mamba models is an average 7 and 3.3 times smaller in mean divergence than Pythia and OpenELM models, respectively**. For Winogrande evaluations, `LIMA` fine-tuning with Mamba models is an average 4.1 and 3.3 times smaller in mean divergence than Pythia and OpenELM models, respectively. **Across the two fine-tuning datasets, two benchmarks, and six evaluated MPFT/PEFT combinations, Pythia and OpenELM produce 9 and 4 large deviation spikes, respectively, while Mamba produces zero.**

### 4.3 Hyperparameter tuning robustness of Mamba models

We empirically demonstrate that Mamba models are robust to the choice of PEFT hyperparemters. Using the training recipe of Tunstall et al. (2023) as a basis, we conduct an extensive hyperparameter search across the learning rate, LoRA dimension, and number of warmup steps. From the Cartesian-product of these three parameters, 150 hyperparameter configurations were sampled and used to fine-tune `Mamba 370M` over the Openhermes dataset (Teknium, 2024), which consists of 242,000 supervised samples. For comparison, `Pythia 410M` is similarly fine-tuned using the same set of 150 hyperparameter configurations.

The MMLU 5-shot performance for each of the 150 Mamba and Pythia fine-tuned models is displayed in Figure 3. `Pythia 410M` is capable of higher performance than `Mamba 370M`, where the average accuracy for the former and the latter are 26.5% and 24.8%, respectively. However, `Mamba 370M` is much more robust to the choice of hyperparameters, with a difference of 1.5% between the minimum (23.3%) and maximum

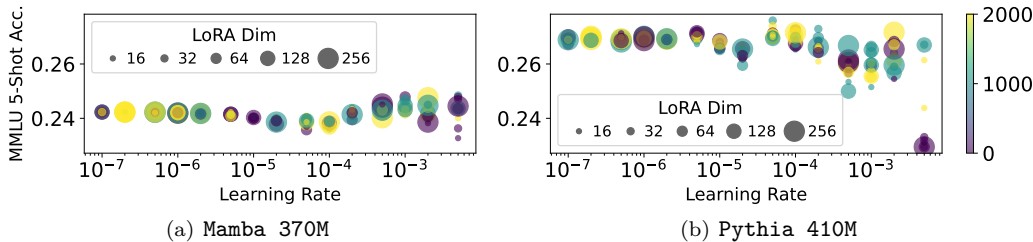

Figure 3: Fine-tuning hyperparameter search for OpenHermes. Each point is a different hyperparameter configuration. `SLL LoRA` was used for both models. The $x$-axis is the learning rate, the $y$-axis is resulting MMLU 5-shot performance, bubble size is the LoRA dimension, and the color is the number of warmup steps $\in \{0, 1\text{k}, 2\text{k}\}$.

(24.8%). In contrast, `Pythia 410M` fine-tuned models display a large performance difference of 4.7% between the minimum (22.9%) and maximum (27.6%).

## 4.4 Stable instruction tuning improves Mamba ICL for natural language tasks

With both stable MPFT and PEFT, we determine the impact of instruction tuning Mamba and Mamba-2 models on ICL for natural language tasks. We instruction fine-tuned Mamba and Mamba-2 pretrained models using `ALL LoRA`, the OpenHermes dataset, and the training recipe described in Section 4 (with `BF16`). Zero and few-shot performance are evaluated using the same five standard natural language benchmarks used in Section 4.3. ICL performance is reported as the *average improvement percentage* of $\{1, 3, 5\}$-*shot* versus 0-*shot* (AIPSS). For comparison, Pythia pretrained models are instruction fine-tuned using the same training recipe and `ALL LoRA` (i.e., all Pythia linear layers are adapted).

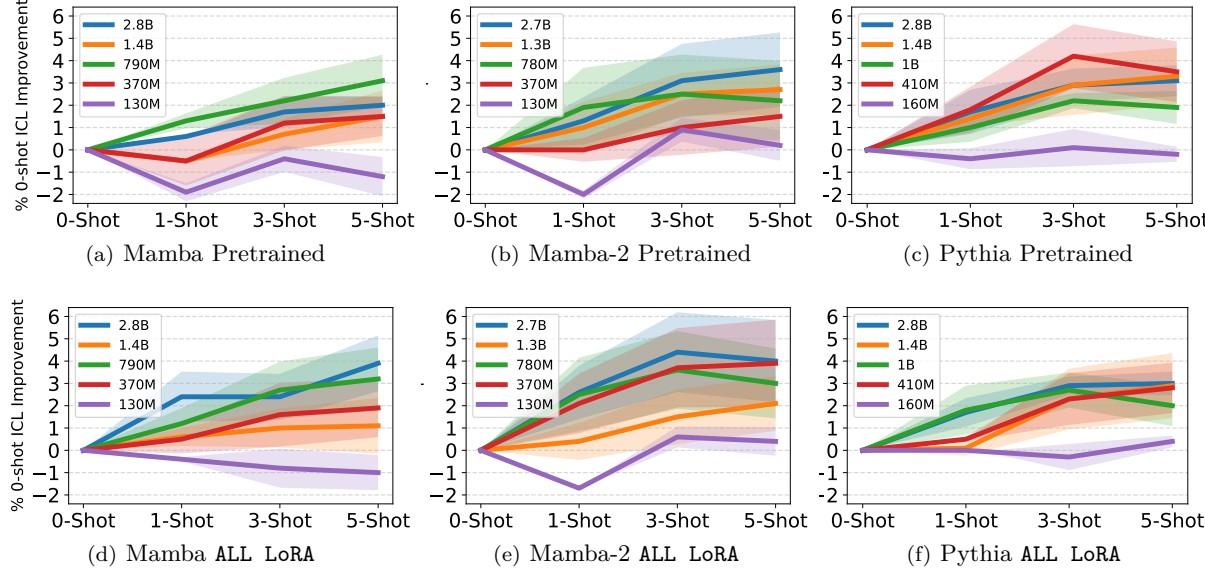

Figure 4: Instruction tuning narrows the ICL gap between Mamba and Pythia, and creates a gap from Pythia to Mamba-2 models. `ALL LoRA` models were instruction tuned on the OpenHermes (Teknium, 2024) dataset for one epoch. Performance is reported as the average improvement percentage of $\{1, 3, 5\}$-shot versus 0-shot over five standard natural language benchmarks.

Figure 4 displays AIPSS for pretrained and instruction fine-tuned Mamba and Pythia models. As previously noted, pretrained Mamba models do not display similar ICL ability as comparable Pythia models on the

evaluated standard NLP benchmarks. In particular, `Mamba 2.8B`, the largest pretrained Mamba model, displays inconsistent zero-shot improvements as the number of shots increase. While pretrained Mamba-2 models display significantly better ICL ability than Mamba models, Mamba-2 models smaller than 780 million parameters struggle.

However, after instruction tuning, all Mamba models larger than `Mamba 130M` consistently improve in ICL performance as the number of shots increase. Similarly, the majority of Mamba-2 models larger than `Mamba-2 130M` greatly improve in ICL performance. Thus, while pretrained Mamba and Mamba-2 models are only capable of 38% and 82% (respectively) of the AIPSS compared to similar pretrained Pythia models, instruction tuned Mamba and Mamba-2 models are capable of 81.5% and 133% of the AIPSS relative to similarly fine-tuned Pythia models.

In addition to Pythia, we compare instruction tuned Mamba models' ICL abilities to other state-of-the Transformer LLMs of comparable sizes and pretraining token counts: `OpenELM` (Mehta et al., 2024) (sizes `270M`, `450M`, and `1.1B`), `TinyLlama 1.1B` (Zhang et al., 2024), and `OLMo 1.2B` (Groeneveld et al., 2024). We additionally group models by parameter count, to understand ICL as an emergent behavior of fine-tuned Mamba models. To further explore the emergence of this ability for Mamba SSMs, we include recent Mamba-specific PEFT methods (Yoshimura et al., 2025). Displayed in Appendix G, pretrained SSMs and Transformers of parameter counts 270 million and less display slight or detrimental ICL abilities (i.e., few-shot performance is worse than zero-shot). For models of greater than 450 million parameters, the majority of SSMs and Transformers display positive ICL abilities, with the exception of `Mamba 1.4B`.

Instruction tuning greatly smooths ICL performance across both parameter classes. Thus, while instruction tuned SSMs and Transformers of 160 million parameters or fewer continue to display slight or detrimental ICL abilities, **all models of 270 million parameters and greater show positive ICL abilities, with `Mamba-2 2.7B` closely matching (or exceeding) the ICL ability of the top performing Transformer LLM per-shot**. Furthermore, in terms of an emergent ability (Wei et al., 2022), ICL emerges for instruction tuned Mamba and Mamba-2 SSMs of size 370 million and greater, while no clear trend presents itself for pretrained models.

## 5 Conclusions and Future Directions

Using dynamical systems theory, we've shown that the recurrent dynamics of Mamba SSMs are stable to small input perturbations. We've extensively confirmed this result, showing that Mamba training is significantly more robust to changes introduced during MPFT and PEFT than Transformer-based LLMs across widely-used fine-tuning datasets and benchmarks. Furthermore, by including many more few-shot evaluations in our divergence metric than previous work, our stability studies uncovered that Transformer LLMs are susceptible to large deviation spikes during MPFT alone, as well as when MPFT is combined with PEFT. As MPFT and PEFT are two of the most common LLM fine-tuning frameworks, and as lower precision formats (e.g., `NF4` and `FP8`) become more widely used for fine-tuning, we advocate for more extensive stability analysis (e.g., including several few-shot evaluations when calculating divergence) to expose large deviation spikes.

In order to mitigate large deviation spikes, we've shown that, unlike Transformer LLMs, Mamba models are extremely stable and do not exhibit this phenomena. Furthermore, in addition to stable MPFT and PEFT, we've shown that a core LLM reasoning ability (ICL) of Mamba models significantly improves with instruction tuning; i.e., we've shown that while pretrained Mamba models' ICL capabilities lag behind those of pretrained LLMs on natural language tasks, instruction tuned Mamba models' rival (and surpass, in many Mamba-2 cases) the ICL capabilities of instruction tuned Transformers. Thus, Mamba models are extremely stable learners with the potential to display the ICL capabilities of attention-based models once instruction tuned.

There are several avenues for future work. In particular, adapting Mamba's CUDA kernels to support more aggressive low-precision PEFT methods (Dettmers et al., 2024) would further decrease the hardware needed to train Mamba models, while providing additional speedups and testing the limits of the derived stability results. Furthermore, our theoretical contributions open the door for several follow up studies. E.g., additional results building off Theorems 3 and 4 may tackle generalization error bounds and privacy fine-tuning (by

considering a language modeling framework and calculating upper bounds under distributions of $\varepsilon$), and Mamba-specific decoding schemes (where new algorithms may exploit the derived deviation bounds ensure next-token predicitions lie within a deviation tolerance from the true optimal decoding).

## 6 Relations to Previous Work

**MPFT divergence for LLMs**. Previous work has studied the performance difference for attention-based LLMs under low-precision inference (Dettmers et al., 2022; Dettmers & Zettlemoyer, 2023) and fine-tuning (Dettmers et al., 2024). However, these works only measured divergence using single-shot performance (i.e., zero-shot in (Dettmers et al., 2022; Dettmers & Zettlemoyer, 2023), 5-shot for MMLU and zero-shot otherwise in (Dettmers et al., 2024)). In contrast, we report the difference in model performance as the mean divergence between the full-precision and mixed-precision models, averaged over the accuracy of four different few-shot evaluations. Thus, in contrast to previous work, our inclusion of multiple shots: a) uncovers significant divergence in widely used LLMs even at widely used mixed-precisions (`FP16` and `BF16`), and (b) more rigorously tests mixed-precision training's effect on one of LLMs' most important reasoning abilities (ICL). We are not aware of any previous works which have attempted to understand the performance impact of MPFT on Mamba models.

**Lyapunov stability and RNNs/SSMs**. Lyapunov exponents have previously been considered (Mikhaeil et al., 2022; Vogt et al., 2022) for classic RNN structures (e.g., vanilla RNNs, LSTMs, GRUs, PLRNNs, etc.), to determine when such models exhibit chaotic dynamics and the impact on the exploding/vanishing gradient phenomena[1].

For S4 (Gu et al., 2022) SSMs, Goel et al. (2022) used Hurwitz matrices to characterize the numerical stability of linear time-invariant (LTI) S4 models. Similarly, Orvieto et al. (2023) used the LTI property to derive the eigenvalue decomposition of S4-like models, providing stability conditions by bounding the corresponding eigenvalues (further discussed in Appendix E). However, such analysis is not applicable to time-varying models, such as Mamba, nor does it characterize the effects of sensitive dependence on initial conditions (e.g., divergence of two $\varepsilon$ close inputs). To the best of our knowledge, no previous works have used Lyapunov exponents to explore the effects of mixed-precision on recurrent neural models or Mamba architectures.

**In-context learning**. Recent work has sought to understand the in-context learning (ICL) capabilities of Mamba LLMs when trained from scratch for specific tasks (Park et al., 2024; Lee et al., 2024). Another line of recent work has sought to understand how hybrid Mamba-Transformer models may be directly distilled from Transformer models (Wang et al., 2024b). However, to the best of our knowledge, no existing works have either theoretically explored the effects small input changes (e.g., due to mixed-precision) have on Mamba's recurrent dynamics or empirically explored such effects downstream impact on fine-tuning.

As previously noted, recent works (Park et al., 2024; Lee et al., 2024) have studied Mamba's ability to perform ICL by training Mamba models for specific tasks. Such tasks include logistic regression, decision trees, and learning other simple function classes, following the work of Garg et al. (2022). We emphasize that, in this set up, relatively small Mamba models–33 million and 90 million parameters for Lee et al. (2024) and Park et al. (2024), respectively–are trained from scratch for every evaluated task. Indeed, Park et al. (2024) notes that subsequent work is necessary to understand Mamba's ICL capabilities for language modeling using standard natural language benchmarks, as well as for larger model sizes. Thus, our study of both the pretrained and instruction tuned ICL capabilities of Mamba/Mamba-2 LLMs for natural language tasks are complimentary to previous works.

## Failure Cases and Limitations

While we explored the use of LoRA for Mamba models, many other PEFT adapters exist (Liu et al., 2022; Li & Liang, 2021; Houlsby et al., 2019; Lester et al., 2021). Furthermore, while mixed-precision using `FP16` and

---

[1]We note that this continues a long line of research exploring RNNs sensitivity to initial conditions and their subsequent ability to produce chaotic output (Ribeiro et al., 2020; Laurent & von Brecht, 2017; Bertschinger & Natschläger, 2004; Bertschinger et al., 2004), although previous work did not leverage Lyapunov exponents.

`BF16` were explored, lower-precision methods exist (Dettmers et al., 2024). Both are interesting directions for future work. Finally, our timing and memory usage experiments using Alpaca did not consider the largest two Mamba models (1.4B and 2.8B) due to their exceeding A10G memory capacity for `FP32` full fine-tuning.

## Broader Impact Statement

The Mamba models considered are all LLMs, and thus have the same potential positive and negative societal impacts as other LLMs (e.g., hallucinations). Furthermore, fine-tuning is known to possibly erode existing LLM guardrails, and thus the described methods may be adapted for this fine-tuning use case (as is the case for all PEFT and MPFT methods). However, MPFT/PEFT is shown to improve the quality of Mamba models for downstream applications, which may be adapted for all positive LLM applications in society (e.g., personal assistants, task automation, code completion, etc.). Finally, the described methods decrease the computational constraints required to train and inference Mamba SSMs, which has implications for green ML (e.g., decreased CO2 emissions, positive climate change impact, etc.).

## Acknowledgments

We thank Leidos for funding this research through the Office of Technology. This manuscript has been approved for public release **24-LEIDOS-0515-27826**. We thank (NeurIPS) Reviewer Uruo, (TMLR) Reviewer z2sE, and (TMLR) Reviewer SpRc for helpful comments and feedback, which spurred the authors to derive and add several additional theoretical results. This work is dedicated to the memory of Bella Halloran; thank you for supporting this one final paper, you are missed.

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

# A    Mamba stable dynamics proof

Recall the state-space parameters and equations for the `MambaBlock`; $\mathbf{A}, \mathbf{B}_t, \mathbf{C}_t, \boldsymbol{\Delta}_t \in \mathbb{R}^{d \times d}$ for $t \in \{1, \ldots, T\} = [T]$. Given an input sequence $\mathbf{u}_1, \ldots, \mathbf{u}_T \in \mathbb{R}^d$, the following linear mapping through latent states $\boldsymbol{x}_1, \ldots, \boldsymbol{x}_T \in \mathbb{R}^d$ is used to produce the output $\mathbf{y}_1, \ldots, \mathbf{y}_T \in \mathbb{R}^d$:

$$
\begin{aligned}
\boldsymbol{x}_t &= \bar{\mathbf{A}}_t \boldsymbol{x}_{t-1} + \bar{\mathbf{B}}_t \mathbf{u}_t \\
\mathbf{y}_t &= \mathbf{C}_t \boldsymbol{x}_t,
\end{aligned}
\tag{4}
$$

where $\bar{\boldsymbol{\Delta}}_t = \texttt{softplus}(\texttt{Linear}(\boldsymbol{\Delta}_t)) \in \mathbb{R}_{\geqslant 0}{}^{d \times d}$, $\bar{\mathbf{A}}_t = \exp(\bar{\boldsymbol{\Delta}}_t \mathbf{A})$, $\bar{\mathbf{B}}_t = \mathbf{A}^{-1}(\bar{\mathbf{A}} - \mathbf{I})\mathbf{B}_t$, and $\mathbb{R}_{\geqslant 0}$ is the set of non-negative real numbers. In practice, $\mathbf{A}, \mathbf{B}_t, \mathbf{C}_t$ and $\boldsymbol{\Delta}_t$ are diagonal matrices.

Furthermore, recall the following definitions:

$$
\boldsymbol{x}_t = F_\theta(\boldsymbol{x}_{t-1}, \mathbf{u}_t)
$$

where $\theta$ denotes the aforementioned time-varying parameters. For input sequence $\mathbf{u}_t, \ldots, \mathbf{u}_T$ and initial latent state value $\boldsymbol{x}_0$, we thus write

$$
\boldsymbol{x}_T = F_\theta(F_\theta(\ldots F_\theta(\boldsymbol{x}_0, \mathbf{u}_1))) \coloneqq F_\theta^{T-1}(\boldsymbol{x}_0, \mathbf{u}_1).
$$

We first prove that, given two scalar $\varepsilon$-close inputs to a `MambaBlock`, their deviations do not grow exponentially as the number of recurrences increases (Lemma 1). The main result in the paper is subsequently proved.

**Lemma 1.** *For input $(\boldsymbol{x}_0, \mathbf{u}_1)$ to a MambaBlock, small changes $(\boldsymbol{x}_0 + \varepsilon, \mathbf{u}_1 + \varepsilon)$ produce deviations which are exponentially non-increasing over discrete-time. That is, $\max |F_\theta^N(\boldsymbol{x}_0, \mathbf{u}_1) - F_\theta^N(\boldsymbol{x}_0 + \varepsilon, \mathbf{u}_1 + \varepsilon)| \in \mathcal{O}(|\varepsilon| \exp(N\zeta))$, for some scalar $\zeta \leqslant 0$.*

*Proof.* Firstly, we note that within the `MambaBlock`, $\mathbf{A}$ is stored in log-space followed by a negative exponentiation prior to use. Thus, $\mathbf{A} \in \mathbb{R}_{\leqslant 0}{}^{d \times d}$, where $\mathbb{R}_{\leqslant 0}$ is the set of non-positive real numbers.

Recall that for the maximum deviation, we have:

$$
\max |F_\theta^N(\boldsymbol{x}_0, \mathbf{u}_1) - F_\theta^N(\boldsymbol{x}_0 + \varepsilon, \mathbf{u}_1 + \varepsilon)| \in \mathcal{O}(|\varepsilon| \exp(N \lambda_{\texttt{max}})).
$$

where the maximal Lyapunov exponent $\lambda_{\texttt{max}}$ is defined as:

$$
\lambda_{\texttt{max}} \coloneqq \lim_{T \to \infty} \frac{1}{T} \log \left\| \prod_{t=0}^{T} \frac{\partial \boldsymbol{x}_t}{\partial \boldsymbol{x}_{t-1}} \right\|_2,
$$

and $\| \| _2$ denotes the spectral norm for matrices.

Thus, to complete the proof, it suffices to show that $\lambda_{\texttt{max}} \leqslant 0$. Recall that $\mathbf{A}$ and $\bar{\boldsymbol{\Delta}}_t$ are diagonal. From Equation 4, we thus have

$$
\begin{aligned}
\lambda_{\texttt{max}} &= \lim_{T \to \infty} \frac{1}{T} \log \left\| \prod_{t=0}^{T} \frac{\partial \boldsymbol{x}_t}{\partial \boldsymbol{x}_{t-1}} \right\|_2 \\
&= \lim_{T \to \infty} \frac{1}{T} \log \left\| \prod_{t=0}^{T} \exp(\bar{\boldsymbol{\Delta}}_t \mathbf{A}) \right\|_2 \\
&= \lim_{T \to \infty} \frac{1}{T} \log \left\| \exp \sum_{t=0}^{T} (\bar{\boldsymbol{\Delta}}_t \mathbf{A}) \right\|_2
\end{aligned}
$$

Let $i$ be the dimension which corresponds to the output of the spectral norm, i.e., $i = \text{argmax}_{j=1,\dots,d}\{\exp\sum_{t=0}^{T}(\bar{\mathbf{\Delta}}_t[j,j]\mathbf{A}[j,j])\}$. We thus have

$$
\begin{aligned}
\lambda_{\texttt{max}} &= \lim_{T\to\infty}\frac{1}{T}\log\left\|\exp\sum_{t=0}^{T}(\bar{\mathbf{\Delta}}_t\mathbf{A})\right\|_2 \\
&= \lim_{T\to\infty}\frac{1}{T}\log\exp\sum_{t=0}^{T}(\bar{\mathbf{\Delta}}_t[i,i]\mathbf{A}[i,i]) \\
&= \mathbf{A}[i,i]\lim_{T\to\infty}\frac{1}{T}\sum_{t=0}^{T}\bar{\mathbf{\Delta}}_t[i,i]
\end{aligned}
$$

$\mathbf{A}[i,i]$ is non-positive and $\lim_{T\to\infty}\frac{1}{T}\sum_{t=0}^{T}\bar{\mathbf{\Delta}}_t[i,i]\geqslant 0$, since $\bar{\mathbf{\Delta}}_t[i,i]\in\mathbb{R}_{\geqslant 0}\;\forall t$. Thus, $\lambda_{\texttt{max}}\leqslant 0$. $\qquad\square$

**Theorem.** *Let $(\boldsymbol{x}_{t-1},\mathbf{u}_t)$ be the latent state and input at an arbitrary time $t\in[1,T]$ within a `MambaBlock`. Then small changes $(\boldsymbol{x}_{t-1}+\varepsilon,\mathbf{u}_t+\varepsilon)$ produce deviations which are exponentially decreasing over discrete-time, i.e., $\max|F_\theta^N(\boldsymbol{x}_0,\mathbf{u}_1)-F_\theta^N(\boldsymbol{x}_0+\varepsilon,\mathbf{u}_1+\varepsilon)|\in\mathcal{O}(|\varepsilon|\exp(N\zeta))$, for some scalar $\zeta\leqslant 0$.*

*Proof.* Let $\tau(t)$ be a function that maps time values such that $\tau(t)\in[1,T-t]$ and $\tau(t)=1,\tau(t+1)=2,\dots,\tau(t+T)=T-t$. Then $\mathbf{B}_{\tau(t)},\mathbf{C}_{\tau(t)},\mathbf{\Delta}_{\tau(t)}$ define a new `MambaBlock` with inputs $\mathbf{u}_{\tau(t)},\dots,\mathbf{u}_{\tau(t+T)}$ and subsequent recurrent states $\boldsymbol{x}_{\tau(t)},\dots,\boldsymbol{x}_{\tau(t+T)}$. Applying Lemma 1 to this `MambaBlock` with $(\boldsymbol{x}_{\tau(t)-1},\mathbf{u}_{\tau(t)})$ completes the proof. $\qquad\square$

## B  Mamba stable outputs proof

**Theorem.** *Assume $(\boldsymbol{x}_{t-1}+\varepsilon,\mathbf{u}_t+\varepsilon)$ produce deviations which are exponentially non-increasing over discrete-time. Then small changes to the output $\mathbf{y}_t$ are also exponentially non-increasing over discrete time.*

*Proof.* Recall that $\boldsymbol{x}_T=F_\theta^T(\boldsymbol{x}_0,\mathbf{u}_1)$. Furthermore, recall from Equations 1 and 2, $\mathbf{y}_t=\mathbf{C}_t\boldsymbol{x}_t$, where $\mathbf{C}_t$ is diagonal.

Let

$$
\mathbf{y}_T=G_\theta^T(\boldsymbol{x}_0,\mathbf{u}_1)=\mathbf{C}_T\boldsymbol{x}_T=\mathbf{C}_T F_\theta^T(\boldsymbol{x}_0,\mathbf{u}_1).
$$

Consider $\varepsilon$-close inputs $(\boldsymbol{x}_{t-1},\mathbf{u}_t)$ and $(\boldsymbol{x}_{t-1}+\varepsilon,\mathbf{u}_t+\varepsilon)$, and their respective outputs $\mathbf{y}_t$ and $\mathbf{y}_t'$. Assume $(\boldsymbol{x}_{t-1}+\varepsilon,\mathbf{u}_t+\varepsilon)$ produce deviations which are exponentially non-increasing over discrete-time. That is, $\max|F_\theta^N(\boldsymbol{x}_{t-1},\mathbf{u}_t)-F_\theta^N(\boldsymbol{x}_{t-1}+\varepsilon,\mathbf{u}_t+\varepsilon)|\in\mathcal{O}(|\varepsilon|\exp(N\zeta))$, for some scalar $\zeta\leqslant 0$.

We thus have

$$
\begin{aligned}
\max|\mathbf{y}_t-\mathbf{y}_t'| &= \max|G_\theta^N(\boldsymbol{x}_{t-1},\mathbf{u}_t)-G_\theta^N(\boldsymbol{x}_{t-1}+\varepsilon,\mathbf{u}_t+\varepsilon)| \\
&= \max|\mathbf{C}_N F_\theta^N(\boldsymbol{x}_{t-1},\mathbf{u}_t)-\mathbf{C}_N F_\theta^N(\boldsymbol{x}_{t-1}+\varepsilon,\mathbf{u}_t+\varepsilon)| \\
&\propto \max|F_\theta^N(\boldsymbol{x}_{t-1},\mathbf{u}_t)-F_\theta^N(\boldsymbol{x}_{t-1}+\varepsilon,\mathbf{u}_t+\varepsilon)|,
\end{aligned}
$$

where proportionality follows due to the diagonality of $\mathbf{C}_N$ and the vector-absolute value. Thus,

$$
\max|G_\theta^N(\boldsymbol{x}_{t-1},\mathbf{u}_t)-G_\theta^N(\boldsymbol{x}_{t-1}+\varepsilon,\mathbf{u}_t+\varepsilon)|\in\mathcal{O}(|\varepsilon|\exp(N\zeta))
$$

$\qquad\square$

## C  Mamba stable AMP and/or PEFT dynamics proof

For an arbitrary `MambaBlock`, let $\boldsymbol{x}_0, \ldots, \boldsymbol{x}_T$ and $\mathbf{u}_1, \ldots, \mathbf{u}_T$ be the trajectory of latent and input states under full-precision fine-tuning. During a forward pass, automatic mixed-precision (AMP) saves time and memory by computing forward activations in half-precision (`FP16` or `BF16`). During a backward pass, AMP computes gradients in half-precision and up-casts to full-precision prior to updating. MFPT thus introduces changes to the inputs $\mathbf{u}_1, \ldots, \mathbf{u}_T$ (which are passed through a `Swish`) and latent states $\boldsymbol{x}_0, \ldots, \boldsymbol{x}_T$ through $\bar{\boldsymbol{\Delta}}_t$ (which is passed through a `softplus`), up/downcasting of parameter updates, and the changes to the aforementioned changes to the input states.

Potentially in concert with MPFT, PEFT via LoRA induces changes by learning low-rank matrices $\hat{\mathbf{A}}_t, \hat{\mathbf{B}}_t, \hat{\mathbf{C}}_t$

$$\hat{\boldsymbol{x}}_t = (\bar{\mathbf{A}}_t + \hat{\mathbf{A}}_t)\boldsymbol{x}_{t-1} + (\bar{\mathbf{B}}_t + \hat{\mathbf{B}}_t)\mathbf{u}_t$$

$$\hat{\mathbf{y}}_t = (\mathbf{C}_t + \hat{\mathbf{C}}_t)\boldsymbol{x}_t$$

Let $\boldsymbol{x}'_0, \ldots, \boldsymbol{x}'_T$ and $\mathbf{u}'_1, \ldots, \mathbf{u}'_T$ be the trajectory of latent and input states under MPFT and/or PEFT. We thus have the following

**Theorem.** *Let $\epsilon^u_t = \max |\mathbf{u}_t - \mathbf{u}'_t|$, i.e., the maximum, positive-scalar difference between the tth MPFT and full fine-tuning input state. Similarly, let $\epsilon^x_t = \max |\boldsymbol{x}_t - \boldsymbol{x}'_t|$ and $\epsilon^* = \max \bigcup_{t=0}^{T} \{\epsilon^u_t, \epsilon^x_t\}$ (where we define $\mathbf{u}_0 = 0$). We thus have*

$$\max |F_\theta^T(\boldsymbol{x}_0, \mathbf{u}_1) - F_\theta^T(\boldsymbol{x}'_0, \mathbf{u}'_1)| \in \mathcal{O}(\epsilon^* \exp(T\zeta)),$$

*for some scalar $\zeta \leqslant 0$. Thus, differences introduced by AMP and/or PEFT for Mamba models do not exponentially compound over discrete-time.*

*Proof.* Consider the $\boldsymbol{\epsilon}$-vector definition of the Lyapunov exponent: given $\lambda_{\max}, \boldsymbol{\epsilon} \in \mathbb{R}^d$, and two initial values $(\boldsymbol{x}_0, \mathbf{u}_1)$ and $(\boldsymbol{x}_0 + \boldsymbol{\epsilon}, \mathbf{u}_1 + \epsilon)$, the maximum deviation between these points grows as:

$$\max |F_\theta^T(\boldsymbol{x}_0, \mathbf{u}_1) - F_\theta^T(\boldsymbol{x}_0 + \boldsymbol{\epsilon}, \mathbf{u}_1 + \epsilon)| \in \mathcal{O}(\|\boldsymbol{\epsilon}\|_2 \exp(T\lambda_{\max})).$$

where $\|\|_2$ denotes the L2-norm for vectors.

Let $f(\cdot)$ be a function which maps a vector to the set of its elements, e.g., for $\boldsymbol{z} \in \mathbb{R}^d$, $f(\boldsymbol{x}) = \{x_1, \ldots, x_d\}$. Thus, note that for $a^* = \max\{|a| : \forall a \in f(\boldsymbol{\epsilon})\}$, $\|\boldsymbol{\epsilon}\|_2 \leqslant a^*\sqrt{d}$ and $\mathcal{O}(\|\boldsymbol{\epsilon}\|_2 \exp(T\lambda_{\max})) \in \mathcal{O}(a^* \exp(T\lambda_{\max}))$.

Let $\mathcal{E} = \bigcup_{t=0}^{T} \{\mathbf{u}_t - \mathbf{u}'_t, \boldsymbol{x}_t - \boldsymbol{x}'_t\}$ (where we define $\mathbf{u}_0 = 0$), and $\epsilon^* = \max\{|a| : \forall a \in \boldsymbol{\epsilon}, \forall \boldsymbol{\epsilon} \in \mathcal{E}\}$. $\forall \boldsymbol{\epsilon} \in \mathcal{E}$, we thus have

$$\max |F_\theta^T(\boldsymbol{x}_0, \mathbf{u}_1) - F_\theta^T(\boldsymbol{x}_0 + \boldsymbol{\epsilon}, \mathbf{u}_1 + \epsilon)| \in \mathcal{O}(\|\boldsymbol{\epsilon}\|_2 \exp(T\lambda_{\max})) \subseteq \mathcal{O}(\epsilon^* \exp(T\lambda_{\max})),$$

Furthermore, $\{\mathbf{u}_t - \mathbf{u}'_t, \boldsymbol{x}_t - \boldsymbol{x}'_t\} \subseteq \mathcal{E}$, and thus

$$\max |F_\theta^T(\boldsymbol{x}_0, \mathbf{u}_1) - F_\theta^T(\boldsymbol{x}_0 - \boldsymbol{x}_0, \mathbf{u}_1 - \mathbf{u}'_1)| \in \mathcal{O}(\epsilon^* \exp(T\lambda_{\max}))$$

Let $\tau(t)$ be a function that maps time values such that $\tau(t) \in [1, T-t]$ and $\tau(t) = 1, \tau(t+1) = 2, \ldots, \tau(t+T) = T - t$. Then $\mathbf{B}_{\tau(t)}, \mathbf{C}_{\tau(t)}, \boldsymbol{\Delta}_{\tau(t)}$ define a new `MambaBlock` with inputs $\mathbf{u}_{\tau(t)}, \ldots, \mathbf{u}_{\tau(t+T)}$ and subsequent recurrent states $\boldsymbol{x}_{\tau(t)-1}, \boldsymbol{x}_{\tau(t)}, \ldots, \boldsymbol{x}_{\tau(t+T)}$. $\forall t \in \{1, \ldots, T\}$, $\{\mathbf{u}_t - \mathbf{u}'_t, \boldsymbol{x}_t - \boldsymbol{x}'_t\} \subseteq \mathcal{E}$, so that

$$\max |F_\theta^{T-t}(\boldsymbol{x}_{\tau(t)} - 1, \mathbf{u}_{\tau(t)}) - F_\theta^{T-t}(\boldsymbol{x}'_{\tau(t)-1}, \mathbf{u}_{\tau(t)})| \in \mathcal{O}(\epsilon^* \exp((T-t)\lambda_{\max})).$$

.

Thus, we have

$$\max |F_\theta^T(\boldsymbol{x}_0, \mathbf{u}_1) - F_\theta^T(\boldsymbol{x}'_0, \mathbf{u}'_1)| \in \mathcal{O}(\epsilon^* \exp(T\lambda_{\max})),$$

where $\lambda_{\max} \leqslant 0$ by Lemma 1.

$\square$

# D `MambaBlock` Maximum Deviation results

## D.1 $\varepsilon$ varying initial conditions

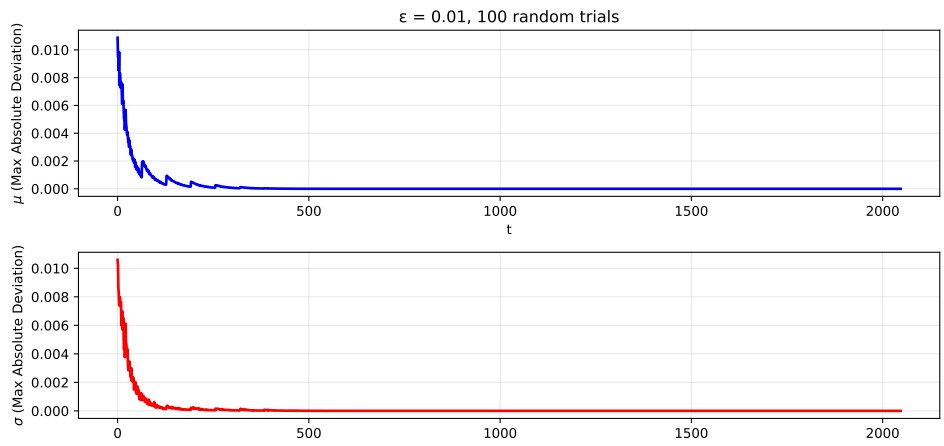

Figure 5: Perturbing the initial states of random `MambaBlock`s produces output state deviations which exponentially decrease over discrete time. $\mu$ denotes the maximum absolute deviation at discrete time $t$ averaged over all trials, while $\sigma$ denotes the respective standard deviation at each discrete time.

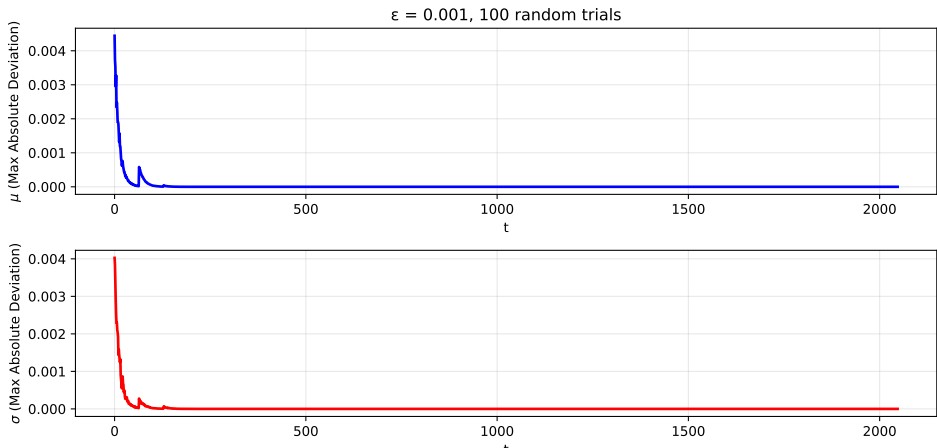

Figure 6: Perturbing the initial states of random `MambaBlock`s produces output state deviations which exponentially decrease over discrete time. $\mu$ denotes the maximum absolute deviation at discrete time $t$ averaged over all trials, while $\sigma$ denotes the respective standard deviation at each discrete time.

# E  Previous SSM stability results

Previous SSM stability results (Goel et al., 2022; Orvieto et al., 2023) consider the following *linear time-invariant* (LTI) equations used in S4 models,

$$\boldsymbol{x}_t = \bar{\mathbf{A}}\boldsymbol{x}_{t-1} + \bar{\mathbf{B}}\mathbf{u}_t \tag{5}$$

$$\mathbf{y}_t = \mathbf{C}\boldsymbol{x}_t., \tag{6}$$

which can be unrolled to produce

$$\boldsymbol{x}_k = \sum_{j=0}^{k} \bar{\mathbf{A}}^j \bar{\mathbf{B}}\mathbf{u}_j. \tag{7}$$

By eigendecomposition,

$$\bar{\mathbf{A}}^j = (\mathbf{P}\boldsymbol{\Lambda}\mathbf{P}^{-1})^j = \mathbf{P}\boldsymbol{\Lambda}^j\mathbf{P}^{-1}.$$

In (Orvieto et al., 2023), the derived sufficient condition for stability is thus $|\alpha_i| \leqslant 1$, for $\boldsymbol{\Lambda} = \mathtt{diag}([\alpha_0, \alpha_1, \ldots, \alpha_d])$. Relatedly, in (Goel et al., 2022), stability is enforced by constraining $\boldsymbol{\Lambda}$ to be Hurwitz, in which case the real part of each diagonal entry $\alpha_i$ is negative, i.e., $\mathrm{Re}(\alpha_i) < 0$, for $\boldsymbol{\Lambda} = \mathtt{diag}([\alpha_0, \alpha_1, \ldots, \alpha_d])$.

Both results critically rely on the LTI property of Equations 5 and 6 to derive Equation 7. However, Mamba's state-space equations are *linear time-varying* (LTV), and thus Equation 7 does not hold. Thus, previous SSM stability results do not directly apply to Mamba models.

# F  Experimental Details

Table 1: Learning rate and LoRA dimension $r$ values

| Mamba size | Mamba-2 size | Pythia size | learning rate | Mamba/Pythia LoRA $r$ | Mamba-2 LoRA $r$ |
|---|---|---|---|---|---|
| 130M | 130M | 160M | 1.0e-5 | 8 | 8 |
| 370M | 370M | 410M | 5.0e-5 | 16 | 16 |
| 790M | 780M | 1B | 1.0e-6 | 32 | 32 |
| 1.4B | 1.3B | 1.4B | 5.0e-6 | 64 | 64 |
| 2.8B | 2.7B | 2.8B | 5.0e-7 | 128 | 64 |

All model checkpoints were evaluated on all benchmarks and few-shot settings using the LM evaluation harness from Eleuther AI (Gao et al., 2023), version `0.4.2`. Pythia and Mamba `Huggingface` checkpoints were used for all inference and fine-tuning experiments, e.g., `EleutherAI/pythia-160m` and `state-spaces/mamba-130m-hf` for the smallest respective models. All fine-tuning experiments were run using package versions `Transformers 4.40.0.dev0`, `Accelerate 0.28.0`, `TRL 0.8.1`, `PyTorch 2.2.1+cu121`, and `PEFT 0.10.0`. All Mamba-2 models were run using `mamba-ssm v2.2.2` using `Huggingface` checkpoints, e.g., `state-spaces/mamba-130m` for the smallest model.

For MPFT, `Flash Attention 2.0` (Dao et al., 2022) via `flash_attn 2.5.7` was used for Pythia models. For `FP16` and `BF16` inference results, Flash Attention 2.0 was used for both Pythia and OLMo models. For OLMo results, the 336B-token checkpoint was used by specifying `revision=step80000-tokens336B`.

All Alpaca and OpenHermes fine-tuning experiments used the following training recipe (adapted from (Tunstall et al., 2023)): `AdamW_torch` optimizer, `cosine annealing` schedule, no gradient accumulation, maximum norm of 1.0 for gradient clipping, and no warmup steps. Training epochs used for all Alpaca and OpenHermes experiments were three and one, respectively. For both Pythia and Mamba models, the learning rate and LoRA dimension $r$ were scaled to improve performance of smaller models (per-model values listed in Table 1).

For `SLL LoRA`, targeted Mamba layers were `{x_proj, embeddings, in_proj, out_proj}`; `x_proj` is a large `MambaBlock` memory buffer which, when targeted by LoRA, adapts parameters $\mathbf{\Delta}_t, \mathbf{B}_t$, and $\mathbf{C}_t$. Pythia targeted `SLL LoRA` layers were `{dense, embed_in, query_key_value, dense_h_to_4h,dense_4h_to_h}`, chosen to balance performance across model sizes.

All experiments were run using a single-GPU Nvidia A10G (24 GB total memory). For Pythia, Mamba, and Mamba-2 `ALL LoRA` experiments in Figure 4, all models followed the same training and PEFT recipes, save for Mamba-2 `2.7B` which required a LoRA $r$ dimension of 64 to fit in A10G memory.

The Alpaca dataset is freely available for download at `https://huggingface.co/datasets/tatsu-lab/alpaca` under open-source license `CC-by-NC 4.0`. The OpenHermes dataset is freely available for download at `https://huggingface.co/datasets/teknium/OpenHermes-2.5` under open-source license `MIT, Apache 2.0, CC`.

Table 2: **Pretrained model performance**. Model checkpoints were evaluated on all benchmarks and few-shot settings using the LM evaluation harness from Eleuther AI (Gao et al., 2023). LAMBADA zero-shot is more effective for the model sizes considered (further discussed in (Xie et al., 2021; Brown et al., 2020)) and thus excluded from few-shot performance averages. Highlighted in bold is the top-performing few-shot learner per benchmark and model grouping.

| Model | N-shot | LAMBADA ppl ↓ | LAMBADA acc ↑ | HellaSwag acc ↑ | PIQA acc ↑ | Arc-E acc ↑ | Arc-C acc ↑ | WinoGrande acc ↑ | 0-shot incr. Mean % ↑ |
|---|---|---|---|---|---|---|---|---|---|
| Mamba 130M | 0 | **16.0** | **44.3** | 35.2 | 64.7 | 48.0 | 24.3 | 52.6 | − |
| | 1 | 19.3 | 38.2 | 35.1 | 64.6 | 47.0 | 23.5 | 50.8 | -1.9 |
| | 3 | 23.1 | 35.2 | 35.0 | 65.1 | 49.1 | 24.0 | 51.0 | -0.4 |
| | 5 | 24.4 | 36.2 | 34.9 | 64.9 | 49.1 | 23.7 | 50.0 | -1.2 |
| Mamba-2 130M | 0 | 16.8 | 43.9 | 35.3 | 64.9 | 47.4 | 24.2 | 52.6 | − |
| | 1 | 20.6 | 37.9 | 34.9 | 64.1 | 46.9 | 23.1 | 51.3 | -2.0 |
| | 3 | 24.3 | 35.1 | 34.9 | 64.4 | **49.0** | 24.7 | **52.9** | **0.9** |
| | 5 | 26.5 | 34.9 | 34.6 | 64.4 | 48.6 | **24.8** | 51.7 | 0.2 |
| Pythia 160M | 0 | 38.2 | 32.7 | 30.2 | 61.8 | 43.4 | 23.8 | 51.0 | − |
| | 1 | 47.2 | 28.2 | 30.6 | 62.2 | 43.4 | 23.7 | 49.3 | -0.4 |
| | 3 | 63.7 | 24.7 | 30.5 | 61.9 | 44.8 | 22.9 | 51.3 | 0.1 |
| | 5 | 66.3 | 25.3 | 30.4 | 62.6 | 43.4 | 23.1 | 50.8 | -0.2 |
| Mamba 370M | 0 | 8.1 | 55.6 | 46.5 | 69.5 | 54.9 | 27.8 | 55.3 | − |
| | 1 | 9.7 | 49.8 | 45.9 | 69.3 | 57.4 | 26.5 | 54.7 | -0.5 |
| | 3 | 10.9 | 48.4 | 46.2 | 69.5 | 58.8 | 28.4 | 53.8 | 1.2 |
| | 5 | 11.4 | 48.6 | 46.2 | 69.4 | 58.3 | 28.0 | **55.9** | 1.5 |
| Mamba-2 370M | 0 | **8.0** | **55.9** | **46.9** | **70.5** | 54.8 | 26.7 | 55.4 | − |
| | 1 | 9.8 | 50.3 | 46.4 | **70.5** | 56.5 | 26.8 | 54.2 | 0.0 |
| | 3 | 11.3 | 48.5 | 46.6 | 70.2 | **59.0** | 26.9 | 54.3 | 1.0 |
| | 5 | 12.5 | 46.6 | 46.7 | 70.3 | 58.5 | 28.2 | 53.3 | 1.5 |
| Pythia 410M | 0 | 10.8 | 51.5 | 40.6 | 66.9 | 52.0 | 24.1 | 53.4 | − |
| | 1 | 12.3 | 47.1 | 40.5 | 68.0 | 53.8 | 25.6 | 52.4 | 1.8 |
| | 3 | 14.4 | 43.2 | 40.9 | 67.9 | 55.1 | 26.9 | 54.0 | **4.2** |
| | 5 | 14.6 | 44.1 | 40.8 | 68.1 | 54.6 | 26.6 | 53.4 | 3.5 |
| Mamba 790M | 0 | 6.0 | 61.4 | **55.1** | 72.3 | 61.2 | 29.5 | 55.9 | − |
| | 1 | 7.1 | 55.9 | 54.5 | 72.4 | 63.0 | 30.2 | 56.9 | 1.3 |
| | 3 | 8.1 | 54.5 | 54.2 | 72.3 | 63.5 | 31.4 | 57.1 | 2.2 |
| | 5 | 8.8 | 52.9 | 54.6 | **72.6** | **64.4** | 31.9 | 57.2 | **3.1** |
| Mamba-2 780M | 0 | **5.9** | **61.7** | 54.9 | 72.0 | 61.0 | 28.5 | **60.2** | − |
| | 1 | 7.1 | 55.5 | 54.7 | 72.4 | 62.3 | 32.1 | 57.1 | 1.9 |
| | 3 | 8.6 | 53.3 | 54.7 | 72.5 | 62.8 | **32.3** | 57.8 | 2.5 |
| | 5 | 9.9 | 51.4 | 55.2 | 72.1 | 62.8 | 32.2 | 56.8 | 2.2 |
| Pythia 1B | 0 | 7.9 | 56.3 | 47.2 | 70.7 | 57.0 | 27.0 | 53.4 | − |
| | 1 | 8.0 | 51.8 | 47.3 | 70.7 | 57.1 | 28.2 | 53.4 | 1.0 |
| | 3 | 10.5 | 48.2 | 47.5 | 71.2 | 59.2 | 28.0 | 54.3 | 2.2 |
| | 5 | 10.9 | 48.4 | 47.3 | 71.4 | 58.7 | 28.4 | 53.1 | 1.9 |
| Mamba 1.4B | 0 | **5.0** | 64.9 | 59.2 | 74.1 | 65.5 | 32.9 | 61.3 | − |
| | 1 | 5.8 | 60.6 | 58.2 | **74.7** | 64.5 | 33.0 | 60.9 | -0.5 |
| | 3 | 6.6 | 58.9 | 58.9 | 73.6 | 66.1 | 34.5 | 60.9 | 0.7 |
| | 5 | 7.0 | 58.3 | 59.0 | 74.1 | 66.4 | 35.5 | 60.4 | 1.5 |
| Mamba-2 1.3B | 0 | **5.0** | **65.6** | 60.0 | 73.2 | 64.2 | 33.1 | 61.1 | − |
| | 1 | 6.0 | 60.1 | 59.4 | 73.1 | 65.6 | 35.3 | 59.4 | 1.0 |
| | 3 | 6.7 | 58.6 | 60.1 | 73.4 | **66.5** | 35.4 | **61.9** | 2.5 |
| | 5 | 7.0 | 58.6 | **60.2** | 73.7 | 66.5 | **35.9** | 61.4 | 2.7 |
| Pythia 1.4B | 0 | 6.1 | 61.7 | 52.1 | 70.9 | 60.5 | 28.5 | 57.4 | − |
| | 1 | 7.0 | 56.3 | 52.1 | 71.4 | 62.0 | 29.5 | 57.5 | 1.4 |
| | 3 | 7.9 | 54.4 | 52.6 | 70.9 | 63.9 | 31.1 | 56.8 | 2.9 |
| | 5 | 8.0 | 54.4 | 52.8 | 71.0 | 63.2 | 31.3 | 57.8 | **3.3** |
| Mamba 2.8B | 0 | 4.2 | 69.1 | 66.1 | 75.2 | 69.6 | 36.4 | 63.3 | − |
| | 1 | 5.0 | 63.7 | 65.6 | 75.6 | 69.9 | 37.1 | 63.9 | 0.6 |
| | 3 | 5.5 | 62.8 | 65.5 | 75.3 | 70.8 | 38.1 | 65.1 | 1.7 |
| | 5 | 5.7 | 62.5 | 66.1 | 76.1 | 70.9 | 38.1 | 64.6 | 2.0 |
| Mamba-2 2.7B | 0 | **4.1** | **69.6** | 66.6 | **76.4** | 69.5 | 36.3 | 63.9 | − |
| | 1 | 4.8 | 65.1 | 65.9 | 75.1 | 70.0 | 38.6 | 65.1 | 1.3 |
| | 3 | 5.3 | 63.9 | 66.8 | 75.2 | **71.9** | 41.0 | 64.1 | 3.1 |
| | 5 | 5.7 | 62.3 | **67.1** | 75.3 | 70.7 | **41.2** | **65.9** | **3.6** |
| Pythia 2.8B | 0 | 5.0 | 64.7 | 59.3 | 73.9 | 64.2 | 32.9 | 59.8 | − |
| | 1 | 5.7 | 60.9 | 59.4 | 73.8 | 66.8 | 34.8 | 59.0 | 1.7 |
| | 3 | 6.2 | 59.1 | 59.9 | 74.7 | 67.4 | 34.9 | 60.8 | 2.9 |
| | 5 | 6.5 | 59.1 | 60.2 | 74.5 | 67.1 | 35.0 | 61.3 | 3.1 |

Table 3: **Instruction tuned model performance**. Model checkpoints were evaluated on all benchmarks and few-shot settings using the LM evaluation harness from Eleuther AI (Gao et al., 2023). LAMBADA zero-shot is more effective for the model sizes considered (further discussed in (Xie et al., 2021; Brown et al., 2020)) and thus excluded from few-shot performance averages. Highlighted in bold is the top-performing few-shot learner per benchmark and model grouping.

| Model | N-shot | LAMBADA ppl ↓ | LAMBADA acc ↑ | HellaSwag acc ↑ | PIQA acc ↑ | Arc-E acc ↑ | Arc-C acc ↑ | WinoGrande acc ↑ | 0-shot incr. Mean % ↑ |
|---|---|---|---|---|---|---|---|---|---|
| Mamba 130M | 0 | **12.9** | **46.5** | **35.1** | 64.2 | 48.7 | **25.5** | 51.7 | – |
| | 1 | 17.8 | 38.1 | 35.0 | 64.2 | 48.6 | 24.9 | **52.2** | -0.4 |
| | 3 | 22.3 | 35.3 | 34.8 | 64.2 | 50.2 | 24.5 | 50.6 | -0.8 |
| | 5 | 23.6 | 35.9 | 34.7 | 64.7 | 49.8 | 24.6 | 50.2 | -0.9 |
| Mamba-2 130M | 0 | 15.2 | 44.5 | **35.1** | 64.5 | 47.2 | 24.7 | **52.2** | – |
| | 1 | 21.9 | 36.1 | 34.5 | 64.3 | 46.8 | 24.0 | 50.8 | -1.7 |
| | 3 | 26.9 | 33.3 | 34.7 | **65.1** | 48.5 | 25.2 | 51.5 | **0.6** |
| | 5 | 29.0 | 33.8 | 34.5 | 64.8 | 48.7 | 25.1 | 51.3 | 0.4 |
| Pythia 160M | 0 | 30.2 | 36.1 | 30.0 | 62.2 | 44.7 | 23.6 | 50.3 | – |
| | 1 | 44.5 | 29.1 | 30.4 | 62.0 | 44.0 | 23.6 | 50.5 | -0.0 |
| | 3 | 66.7 | 25.5 | 30.3 | 62.8 | 45.2 | 22.8 | 49.8 | -0.3 |
| | 5 | 70.4 | 25.3 | 30.5 | 62.9 | 44.1 | 23.4 | 50.8 | 0.3 |
| Mamba 370M | 0 | 7.2 | **56.0** | 46.3 | 69.2 | 55.3 | 27.7 | **56.0** | – |
| | 1 | 9.3 | 49.9 | 45.7 | 68.7 | 57.1 | 28.3 | 55.4 | 0.5 |
| | 3 | 10.4 | 49.4 | 45.7 | 68.9 | 58.7 | **29.7** | 54.1 | 1.6 |
| | 5 | 11.0 | 48.3 | 45.7 | 70.1 | 59.3 | 29.1 | 54.5 | 1.9 |
| Mamba-2 370M | 0 | **7.6** | 54.7 | **46.8** | 69.3 | 52.2 | 27.0 | **56.0** | – |
| | 1 | 9.9 | 48.3 | 46.0 | 69.6 | 55.7 | 28.8 | 55.2 | 2.1 |
| | 3 | 11.8 | 46.3 | 46.3 | 70.1 | 59.0 | 29.1 | 54.5 | 3.6 |
| | 5 | 12.6 | 45.5 | 46.3 | **70.8** | **59.6** | 29.5 | 53.0 | **3.8** |
| Pythia 410M | 0 | 13.3 | 46.4 | 40.9 | 67.4 | 52.7 | 25.4 | 53.4 | – |
| | 1 | 17.2 | 40.4 | 40.5 | 68.4 | 53.6 | 25.7 | 53.0 | 0.5 |
| | 3 | 21.1 | 37.4 | 40.9 | 67.7 | 55.7 | 27.1 | 52.6 | 2.3 |
| | 5 | 21.5 | 38.2 | 40.7 | 67.8 | 55.7 | 27.3 | 53.8 | 2.8 |
| Mamba 790M | 0 | 5.2 | 62.8 | 55.6 | 72.8 | 62.4 | 30.6 | 56.2 | – |
| | 1 | 6.3 | 56.6 | 54.9 | 72.7 | 64.6 | 31.7 | 56.3 | 1.2 |
| | 3 | 7.0 | 55.6 | 54.7 | 72.4 | **65.3** | 33.2 | 57.5 | 2.7 |
| | 5 | 7.5 | 54.6 | 54.9 | 72.9 | 65.6 | 33.8 | 57.2 | 3.2 |
| Mamba-2 780M | 0 | **4.9** | **63.4** | **55.8** | 71.7 | 61.1 | 30.6 | **59.2** | – |
| | 1 | 6.6 | 55.2 | 54.4 | 72.7 | 64.2 | 34.0 | 57.6 | 2.5 |
| | 3 | 7.8 | 52.7 | 54.9 | **73.5** | 65.0 | **34.6** | 57.8 | **3.6** |
| | 5 | 8.6 | 52.8 | 54.8 | 73.4 | 64.6 | 34.0 | 58.0 | 3.1 |
| Pythia 1B | 0 | 7.7 | 56.6 | 47.3 | 70.8 | 57.1 | 26.7 | 53.4 | – |
| | 1 | 8.8 | 52.0 | 47.4 | 70.7 | 57.5 | 28.8 | 53.6 | 1.8 |
| | 3 | 10.2 | 48.7 | 47.5 | 71.4 | 59.0 | 28.5 | 54.4 | 2.6 |
| | 5 | 10.6 | 48.8 | 47.4 | 71.5 | 58.9 | 28.4 | 53.0 | 2.0 |
| Mamba 1.4B | 0 | **4.6** | **64.8** | 59.3 | 74.3 | 65.2 | 35.1 | 62.3 | – |
| | 1 | 5.4 | 60.3 | 58.2 | 74.3 | 66.7 | 35.7 | **62.8** | 0.6 |
| | 3 | 6.1 | 59.3 | 58.4 | 74.1 | 67.4 | 36.6 | 61.8 | 1.0 |
| | 5 | 6.3 | 58.8 | 58.8 | **74.5** | 68.3 | **37.0** | 59.9 | 1.1 |
| Mamba-2 1.3B | 0 | 4.9 | 63.0 | **60.1** | 73.8 | 64.0 | 34.8 | 61.3 | – |
| | 1 | 6.1 | 58.2 | 59.2 | 74.2 | 67.0 | 35.0 | 60.1 | 0.5 |
| | 3 | 7.0 | 56.6 | 59.4 | 73.7 | 67.8 | 36.6 | 59.9 | 1.5 |
| | 5 | 7.2 | 56.5 | 59.9 | 73.5 | **68.5** | 36.7 | 60.7 | 2.2 |
| Pythia 1.4B | 0 | 5.2 | 63.6 | 52.9 | 71.1 | 61.2 | 30.3 | 58.2 | – |
| | 1 | 6.2 | 57.4 | 52.7 | 71.7 | 62.2 | 30.6 | 56.9 | 0.2 |
| | 3 | 7.0 | 56.1 | 53.1 | 71.1 | 64.5 | 32.8 | 56.8 | 2.3 |
| | 5 | 7.1 | 55.5 | 53.3 | 71.2 | 63.8 | 33.5 | 57.5 | **2.9** |
| Mamba 2.8B | 0 | 4.0 | 67.7 | 66.4 | 75.6 | 68.4 | 36.6 | 64.2 | – |
| | 1 | 4.8 | 63.3 | 65.9 | 76.2 | 70.9 | 39.4 | 64.6 | 2.4 |
| | 3 | 5.3 | 62.1 | 65.7 | 75.8 | 71.3 | 39.1 | 65.4 | 2.4 |
| | 5 | 5.4 | 61.9 | 66.2 | 77.2 | 71.4 | 40.4 | 66.1 | 3.9 |
| Mamba-2 2.7B | 0 | **3.8** | **68.4** | **67.5** | 76.0 | 69.5 | 38.3 | 65.3 | – |
| | 1 | 4.5 | 63.8 | 66.7 | 76.0 | 71.8 | 41.5 | **67.1** | 2.6 |
| | 3 | 5.0 | 62.3 | 67.3 | 76.2 | **73.3** | 44.4 | 66.0 | 4.5 |
| | 5 | 5.3 | 61.8 | 67.4 | **76.4** | 72.4 | **44.5** | 65.0 | **4.1** |
| Pythia 2.8B | 0 | 5.0 | 64.7 | 59.3 | 74.0 | 64.7 | 33.3 | 59.2 | – |
| | 1 | 5.6 | 60.8 | 59.5 | 74.0 | 66.7 | 34.9 | 59.3 | 1.7 |
| | 3 | 6.1 | 59.2 | 59.9 | 75.0 | 67.5 | 34.9 | 60.9 | 2.9 |
| | 5 | 6.5 | 59.0 | 60.4 | 74.5 | 67.0 | 35.1 | 61.2 | 3.0 |

### F.1 Hardware throughput and memory-utilization improvements given PEFT and MPFT

With stable dynamics and observed divergences smaller than comparable Transformers, we show that MPFT and PEFT may be used to significantly increase GPU-training throughput for Mamba SSMs. To demonstrate such improvements, we utilize the previous fine-tuning settings for the Alpaca dataset. However, we now adjust the batch size to maximize throughput per MPFT and PEFT configuration.

For each MPFT and PEFT configuration, the *average tokens-per-second* (ATPS) is calculated as the total tokens used for fine-tuning divided by total training time, and the *maximum memory-per-token* (MMPT) is calculated as the maximum GPU memory utilization incurred (over the entire fine-tuning run) divided by the total number of tokens in each mini-batch. Results are plotted in Figure 7.

Both throughput and memory utilization improve as the number of Mamba parameters increases in Figure 7. **Compared to the full-precision full fine-tuning of `Mamba 790M`** (the largest model supported by an `A10G`'s memory capacity), evaluated **MPFT and PEFT combinations result in an average 2.15 times more training tokens-per-second while reducing per-token memory utilization by an average 62.7%**. Across all model sizes, evaluated MPFT and PEFT combinations result in an average 1.74 times more training tokens-per-second while reducing per-token memory utilization by an average 47.2% compared to respective full-precision fine-tuned runs. Furthermore, while full fine-tuning is no longer possible on a single `A10G` for Mamba models greater than 790 million parameters, MPFT and PEFT allow training Mamba models up to 2.8 billion parameters on GPUs with as little as 24 GB onboard memory.

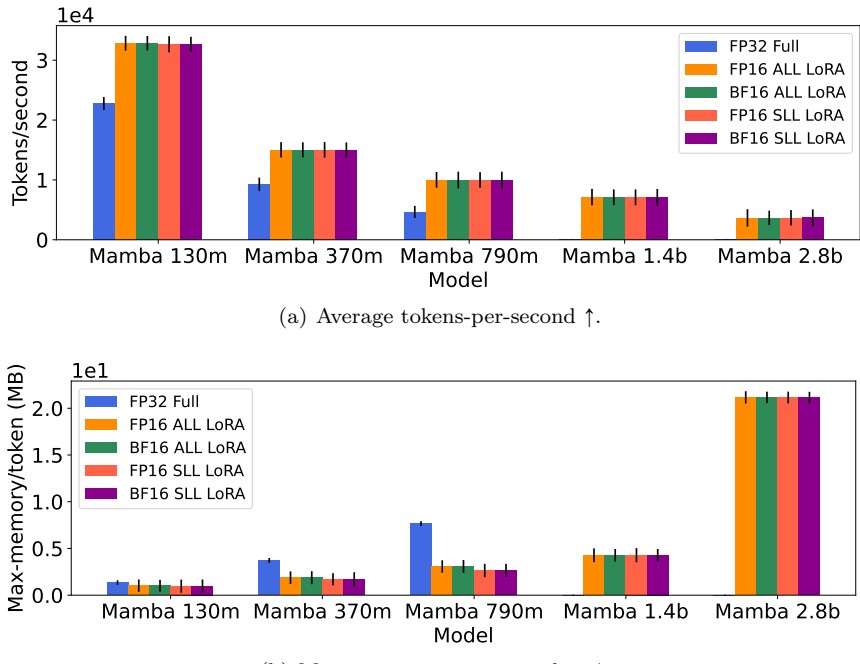

(a) Average tokens-per-second ↑.

(b) Maximum memory-per-token ↓.

Figure 7: Timing and memory usage calculated Mamba model-sizes and PEFT combinations. Each model was trained using the Alpaca dataset dataset for three epochs and maximum sequence length 512. For each PEFT combination, the batch size was tuned to maximize GPU occupancy. Full fine-tuning exceeds available GPU memory (24 GB) for models greater than 790 million parameters.

## G   ICL as an emergent ability of Mamba SSMs

We study the emergent behavior (as a function of model size) of Mamba/Mamba-2 SSMs' ICL abilities on natural language tasks by comparing to a larger number of Transformer-based LLMs of varying sizes. We compare to `OpenELM` (Mehta et al., 2024) (sizes `270M`, `450M`, and `1.1B`), `TinyLlama 1.1B` (Zhang et al.,

2024), and `OLMo 1.2B` (Groeneveld et al., 2024). To limit the emergent effects on both parameter size and pretraining token counts, we did not evaluate models greater than 2.8 billion parameters and chose open-source checkpoints as close as possible to the 300 billion total pretraining tokens used for Mamba, Mamba-2, and Pythia models. Thus, pretraining token counts for OpenELM, TinyLlama, and OLMo models were 429 billion, 503 billion, and 336 billion, respectively. We note this potentially biases ICL performance in favor of the newly evaluated Transformer-based LLMs, and that direct comparisons between Mamba, Mamba-2, and Pythia are the most fair (as these three classes of models were all pretrained on the same dataset for the same number of total pretraining tokens).

We repeat the experiments from Figure 4, where we evaluate the pretrained and instruction tuned ICL capabilities of all models. To understand the critical role of parameter counts, we group all models into two classes: LLMs containing 450 million parameters or less, and LLMs containing greater than 450 million parameters. ICL performance measured by AIPSS is displayed in Figure 8.

From Figure 8, it is clear that pretrained SSMs and Transformers of parameter counts 270 million and less display slight or detrimental ICL abilities (i.e., few-shot performance is worse than zero-shot). For models of greater than 450 million parameters, the majority of SSMs and Transformers display positive ICL abilities, with `Mamba 1.4B` being an outlier in terms of poor performance. With the exception of TinyLlama at 1-shot performance and `Mamba-2 2.7B` for 3- and 5-shot performance, the majority of other pretrained models cluster together.

Instruction tuning greatly smooths ICL performance across both parameter classes. While instruction tuned SSMs and Transformers of 160 million parameters or fewer continue to display slight or detrimental ICL abilities, all parameters of 270 million and greater show positive ICL abilities. For instruction tuned models of greater than 450 million parameters, all SSMs and Transformers show positive ICL abilities, with `Mamba-2 2.7B` greatly outperforming all other models (both SSM and Transformer) in this class.

Thus, in terms of ICL as a function of SSM model size, while no clear trend presents itself for pretrained models, ICL emerges for instruction tuned Mamba and Mamba-2 SSMs of size 370 million and greater.

## H   Expanded divergence results: Alpaca and LIMA fine-tuning, MMLU and Winogrande benchmarks, Mean and Standard Deviation divergences

We extend the non-divergent Mamba fine-tuning results from Section 4. Recall that the following MPFT and PEFT configurations are considered to fine-tune each considered LLM:

1. Full fine-tuning in `FP32`

2. Full fine-tuning in `FP16`

3. Full fine-tuning in `BF16`

4. `ALL LoRA`fine-tuning in `FP32`

5. `ALL LoRA`fine-tuning in `FP16`

6. `ALL LoRA`fine-tuning in `BF16`

7. `SLL LoRA`fine-tuning in `FP32`

8. `SLL LoRA`fine-tuning in `FP16`

9. `SLL LoRA`fine-tuning in `BF16`

In addition to the `Alpaca` dataset (Taori et al., 2023), we also fine-tune all models using the `LIMA` dataset (Zhou et al., 2024). Models are trained using `LIMA` for 5 epochs, while all other settings follow the fine-tuning recipe used for `Alpaca` (described in Appendix F).

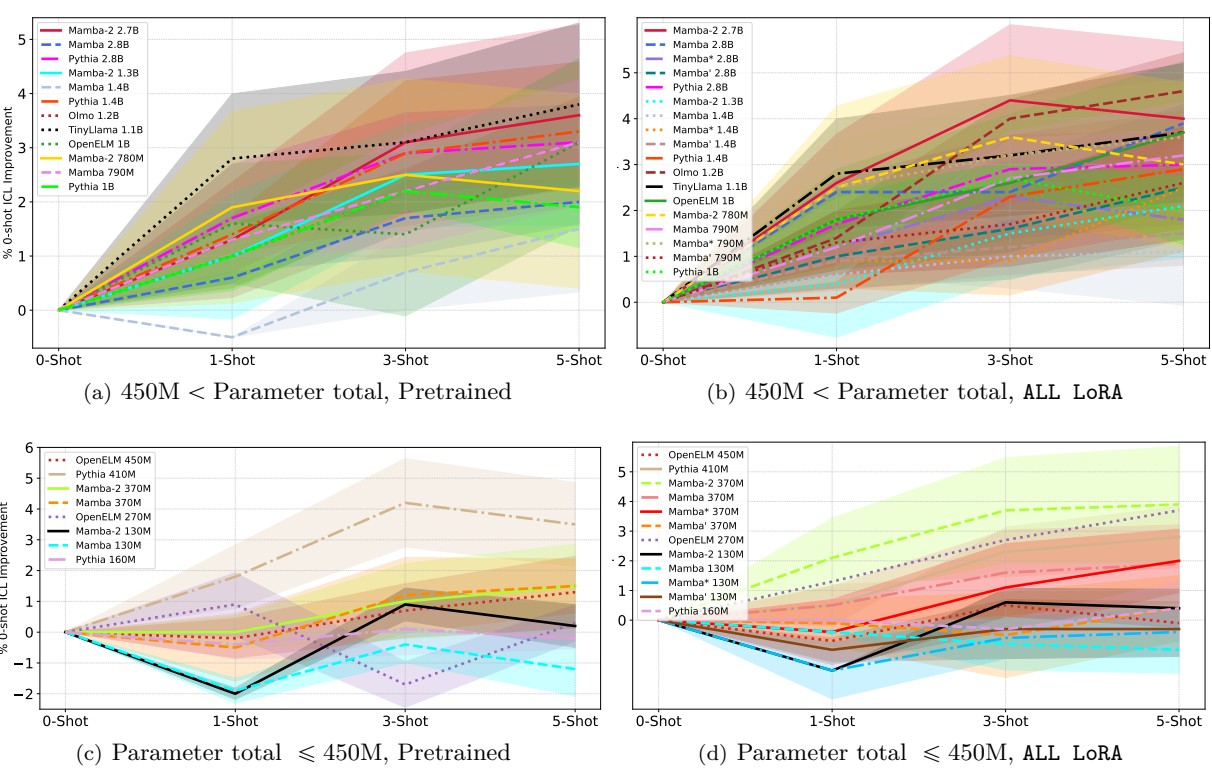

Figure 8: Instruction tuning improves Mamba-2 ICL performance past Transformer LLMs. `ALL LoRA` models were instruction fine-tuned on the OpenHermes dataset for one epoch. Performance is reported as the average improvement percentage of {1, 3, 5}-shot versus 0-shot over five standard natural language benchmarks: HellaSwag, PIQA, Arc-E, Arc-C, and WinoGrande. Mamba* and Mamba' refer to recent Mamba-1 PEFT methods, i.e., $\text{LoRA}_p(X)$ and Additional-Scan, respectively, from Yoshimura et al. (2025).

For natural language benchmarks, in addition to MMLU, we evaluate each fine-tuned model using Winogrande (Sakaguchi et al., 2021). Recall that, for each benchmark, divergence between a mixed-precision fine-tuned model is measured between its full-precision counterpart and averaged over {0, 1, 3, 5}-shot performance. In addition to the average divergence, we also include the standard deviation of divergence. Thus, **in total, 144 LLMs were fine-tuned, 576 MMLU evaluations were conducted, and 576 Winogrande evaluations were conducted.**

Figure H displays results for `Alpaca` and MMLU, Figure H displays results for `Alpaca` and Winogrande, Figure H displays results for `LIMA` and MMLU, Figure H displays results for `LIMA` and Winogrande. Summary statistics for all experiments are presented in Table H. While OpenELM exhibits large deviation spikes for both `Alpaca` benchmark evaluations–and Pythia exhibits large deviation spikes for all four evaluations–**Mamba does not exhibit a single large deviation spike on any benchmark for all considered model sizes and MPFT/PEFT configurations** (i.e., 18 total configurations excluding the full-precision baselines). Furthermore, Mamba models are significantly more stable for MPFT/PEFT compared to Transformer-based LLMs. E.g., **for MMLU evaluations, `Alpaca` fine-tuning with Mamba models is an average 2.6 times smaller in mean divergence than both Pythia and OpenELM models, while `LIMA` fine-tuning with Mamba models is an average 7 and 3.25 times smaller in mean divergence than Pythia and OpenELM models, respectively**.

Table 4: Summary of divergence results for `Alpaca` and `LIMA` fine-tuning datasets, MMLU and Winogrande benchmarks, and Mamba, OpenELM, and Pythia models. For each deviation summary statistic per fine-tuning dataset and benchmark, the lowest deviation is highlighted in bold.

| (Fine-tuning dataset), Benchmark | Architecture | Large deviation spikes ↓ | Avg mean divergence ↓ | Std mean divergence ↓ |
|---|---|---|---|---|
| (`Alpaca`, MMLU) | Pythia | 1 | 0.37 | 0.41 |
| | OpenELM | 1 | 0.37 | 0.32 |
| | Mamba | **0** | **0.14** | **0.08** |
| (`Alpaca`, Winogrande) | Pythia | 4 | 0.72 | 0.58 |
| | OpenELM | 3 | 0.59 | 0.37 |
| | Mamba | **0** | **0.25** | **0.09** |
| (`LIMA`, MMLU) | Pythia | 1 | 0.28 | 0.34 |
| | OpenELM | 0 | 0.13 | 0.15 |
| | Mamba | **0** | **0.04** | **0.03** |
| (`LIMA`, Winogrande) | Pythia | 3 | 0.45 | 0.45 |
| | OpenELM | 0 | 0.36 | 0.18 |
| | Mamba | **0** | **0.11** | **0.12** |

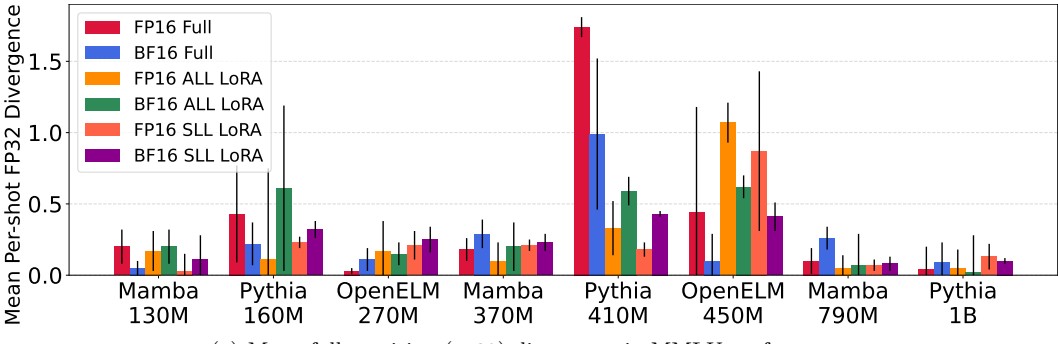

(a) Mean full-precision (`FP32`) divergence in MMLU performance.

Figure 9: **Alpaca fine-tuning, MMLU evaluation.** Mamba, Pythia, and OpenELM models are fine-tuned over the `Alpaca` dataset using different combinations of MPFT and PEFT. Full fine-tuning (i.e., no PEFT adapters) is denoted as `Full`.

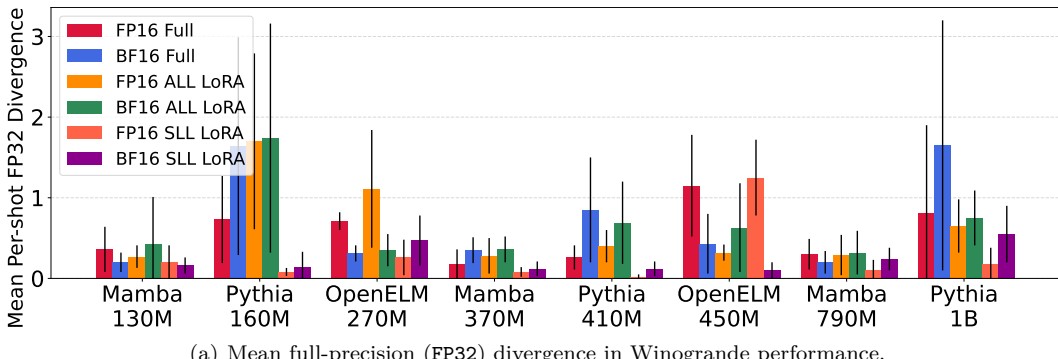

(a) Mean full-precision (`FP32`) divergence in Winogrande performance.

Figure 10: **Alpaca fine-tuning, Winogrande evaluation.** Mamba, Pythia, and OpenELM models are fine-tuned over the `Alpaca` dataset using different combinations of MPFT and PEFT. Full fine-tuning (i.e., no PEFT adapters) is denoted as `Full`.

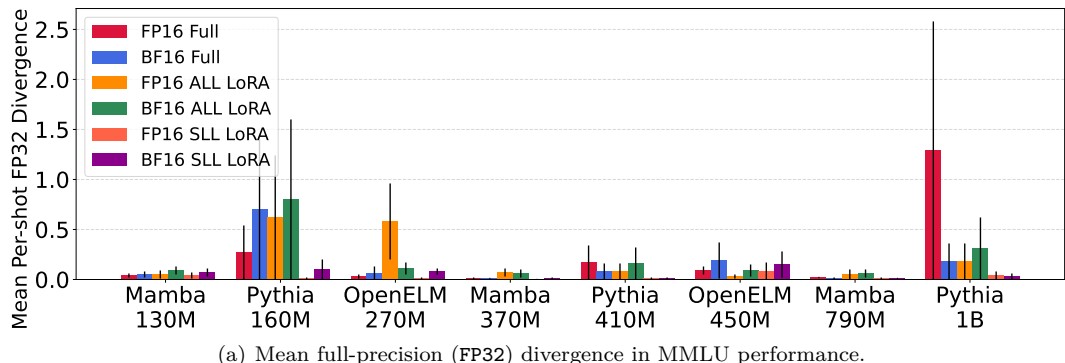

(a) Mean full-precision (`FP32`) divergence in MMLU performance.

Figure 11: **LIMA fine-tuning, MMLU evaluation.** Mamba, Pythia, and OpenELM models are fine-tuned over the `LIMA` dataset using different combinations of MPFT and PEFT. Full fine-tuning (i.e., no PEFT adapters) is denoted as `Full`.

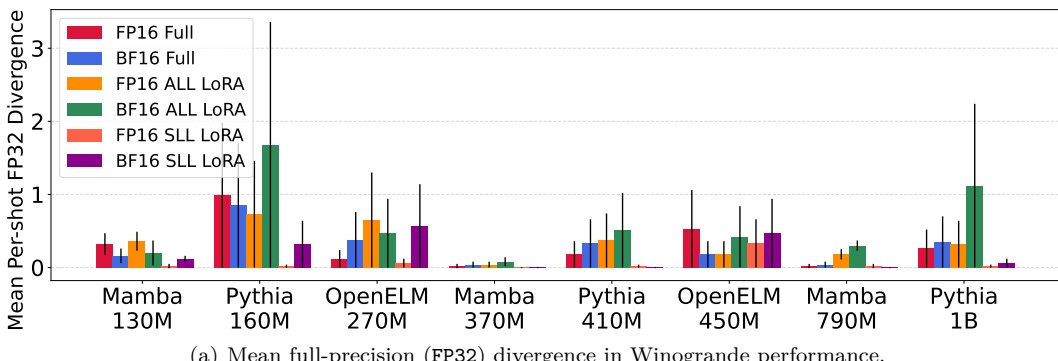

(a) Mean full-precision (`FP32`) divergence in Winogrande performance.

Figure 12: **LIMA fine-tuning, Winogrande evaluation.** Mamba, Pythia, and OpenELM models are fine-tuned over the `LIMA` dataset using different combinations of MPFT and PEFT. Full fine-tuning (i.e., no PEFT adapters) is denoted as `Full`.

