# OpenReview forum: "Mamba State-Space Models Are Lyapunov-Stable Learners"
_TMLR — Accepted by TMLR_

### Review · Reviewer_qxVf · 2025-07-01

**Summary Of Contributions:**

This paper observes that Mamba state space models are Lyapunov-stable, and argues that this provides an explanation for their experiments showing that Mamba models fine-tuned either using full precision, mixed precision, or low-rank updates all behave robustly.

**Audience:**

Yes

**Broader Impact Concerns:**

None, other than that I wonder whether the authors overstate somewhat the implications of their work for reducing emissions.

**Claims And Evidence:**

No

**Requested Changes:**

1. The authors emphasize the fact that previous SSM stability results do not apply to Mamba because it is not an LTI system, and thus put forward a proof of stability based on computation of the maximum Lyapunov exponent. I have two concerns regarding this result (questions 1 and 2 here). First, it risks feeling like it borders on tautological, given that the Mamba designers presumably chose the particular parameterizations that lead to this in order to guarantee stability of the linear dynamics. In particular, a key part of the author's argument for stability is the fact that Mamba chooses a parameterization such that $A_{ii} \leq 0$, which is fundamentally what guarantees stability (see below). I imagine the authors will disagree with this assessment, and I would ask them to elaborate on the reasons why.

2. Second, unless I am mistaken the claim follows directly without the need to invoke the general machinery of Lyapunov stability, as this is still a linear system, time-variant though it may be. As the authors remark, all matrices of interest are in practice chosen to be diagonal, and even if the parameters are time-varying one can write down a formal solution to the dynamics for each component so long as the parameters depend on time (and perhaps the input signal) but not on the hidden state (note that this is consistent with the fact that they write $\partial x\_{t}/\partial x\_{t-1} = \bar{A}\_{t}$). In particular, one is interested in the state dynamics $x\_{t+1} = \bar{A}\_{t} x\_{t} + \bar{B}\_{t} u\_{t}$, which has formal solution $x\_{T+1} = \prod\_{t=0}^{T} \bar{A}\_{t} x\_{0} + \sum\_{t=0}^{T} \prod\_{t'=t+1}^{T} \bar{A}\_{t'} \bar{B}\_{t} u\_{t}$. Here, the order of matrices in the products does not matter given that they are assumed to be diagonal, but one can always fix the convention $\prod\_{t=0}^{T} \bar{A}\_{t} = \bar{A}\_{T} \bar{A}\_{T-1} \cdots \bar{A}\_{0}$. Then, given the assumption that $\bar{A} = \exp(\bar{\Delta}\_{t} A)$ for diagonal matrices $\bar{\Delta}\_{t}$ and $A$, and that $\bar{B}\_{t}$ is likewise diagonal, for each $i \in [d]$ one will have terms like $\prod\_{t=1}^{T} (\bar{A}\_{t})\_{ii} = \exp( A\_{ii} \sum\_{t=1}^{T} (\bar{\Delta}\_{t})\_{ii} )$. Under the assumptions stated in Appendix A (*but not in the main text, this must be rectified*), $A\_{ii} \leq 0$. Then, one has $\prod\_{t=1}^{T} (\bar{A}\_{t})\_{ii} \leq \exp( T \zeta_{i} )$, where $\zeta_{i} = A\_{ii} \min\_{t} (\bar{\Delta}\_{t})\_{ii} \leq 0$, using the assumption that $(\bar{\Delta}_{t})\_{ii} \geq 0$ for all times $t$. One would then get a global sensitivity bound in terms of $\zeta = \max\_{i \in [d]} \zeta\_{i}$ that is exponentially decaying in the time lag since a perturbation was introduced. Can you clarify what I'm missing here - what is the obstacle to this approach if in practice one only needs to consider the diagonal case?

3. The figures in the main text do not include error bars, and standard deviations are reported only in Appendix G in separate bar plots. This makes it very challenging for the reader to assess the statistical significance of results, and as such I ask that the authors update all of their main-text figures to include error bars. This is critical for the legibility of the results, and therefore for acceptance.

4. In equation (4-6), the authors adopt a particular format for storing the parameters in the memory buffer matrix $W$, and then performs low-rank updates to this format of $W$. Though choosing this format is advantageous from the perspective of computational efficiency, it is not clear to me whether this format is in some sense optimal for low-rank fine-tuning, i.e., whether standard matrix rank is an appropriate measure of the complexity of an update given the underlying parameters one fundamentally wants to target. Can you provide more justification here, particularly as it is this formatting that leads to the claimed result that LORA induces dependencies between the different Mamba weights?

5. Can you compare your results to those of [Yoshimura et al. (2025)](https://openreview.net/forum?id=UAKnJMIBwf) on parameter-efficient fine-tuning for Mamba models?

6. A tangential comment - the authors might find [this recent paper by Haiping Huang's group](https://journals.aps.org/pre/abstract/10.1103/PhysRevE.111.034308) on using Lyapunov exponents to measure stability of vanilla RNNs to weight perturbations to be of interest.

**Strengths And Weaknesses:**

The empirical findings of this paper should be of interest to the TMLR audience. However, the way in which they are reported is a weakness of the paper. The authors do not show error bars in their main figures, instead deferring measures of variability to tables and separate bar plots in the Appendices. This makes it challenging to assess the statistical strength of their claim that "training Mamba LLMs is significantly more stable than comparable Transformer-based LLMs". I also have some concerns about the theoretical results presented, as I believe they are amenable to a simpler proof. I elaborate on these and other issues under **Requested Changes** below.

---

> ### Author Response · Authors · 2025-07-14
> **Reply**
>
> Dear Reviewer qxVf,
>
> We thank the reviewer for pointing out that we can combine the mean and standard deviation figures, please find the updated plots in the revised manuscript.
>
> > 1. ... given that the Mamba designers presumably chose the particular parameterizations that lead to this in order to guarantee stability of the linear dynamics.
>
> First, we note this question requests the authors speculate on the intentions of the authors of [1], which we cannot do without breaking double blind.  Furthermore, intentions do not equate to formal proofs; [1] and subsequent works did not formally derive the stability of Mamba SSMs, which is a worthy contribution given the significant interest these models have garnered since their release.
>
> > 2. Second, unless I am mistaken the claim follows directly without the need to invoke the general machinery of Lyapunov stability, as this is still a linear system, time-variant though it may be ...
>
> A "simpler" stability proof is an interesting question.  However, the reviewer's argument is incorrect.  In particular:
> > even if the parameters are time-varying one can write down a formal solution to the dynamics for each component so long as the parameters depend on time (and perhaps the input signal) but not on the hidden state (note that this is consistent with the fact that they write $\partial x_{t}/\partial x_{t-1} = \bar{A}_{t}$)
>
> This is not correct.  As explicitly stated in the manuscript, the quantity $\partial x_{t}/\partial x_{t-1} = \bar{A}_{t}$ **is a part of the definition of the Lyapunov exponent** in Equation 5.  We are able to use this specific form of the Lyapunov exponent since [3] proved the well known result that nonnegativity of the maximal Lyapunov exponent of the system--as defined both in our paper and in [2]--implies Lyapunov stability.  To derive a "simpler" stability result without considering Lyapunov stability, $\bar{B}_t$ must also be accounted for.  However, the entire point of the paper is to characterize how **differences in the input states affect difference in the output states.** This is exactly the definition of Lyapunov stability in Equation 4.  If considering some alternate form of stability, the bounds in Theorem 1, 2, 3, and 4 would not be readily available.
>
> Finally, we ask the reviewer to please look at the proofs for Theorems 1, 2, 3, and 4.  There is much more work to prove the various theoretical results after deriving Lemma 1.
>
> > Under the assumptions stated in Appendix A (but not in the main text, this must be rectified), $A_ii \leq 0$
>
> The reviewer is incorrect, we do not assume this.  This is a direct consequence of the definition of A in the Mamba Block as appropriately stated in the Appendix, i.e.: "Firstly, we note that within the MambaBlock, A is stored in log-space followed by a negative exponentiation prior to use."
>
> > 3. The figures in the main text do not include error bars, and standard deviations are reported only in Appendix G in separate bar plots. This makes it very challenging for the reader to assess the statistical significance of results, and as such I ask that the authors update all of their main-text figures to include error bars. This is critical for the legibility of the results, and therefore for acceptance.
>
> We thank the reviewer for pointing this out, please find the updated plots in the revised manuscript.
>
> > 4. In equation (4-6), the authors adopt a particular format...
>
> We thank the reviewer for the question.  To address other reviewer feedback, we have removed this result in favor of more relevant Lyapunov stability results directly assuming MPFT/PEFT (Theorems 3 and 4).
>
> > 5. Can you compare your results to those of Yoshimura et al. (2025) on parameter-efficient fine-tuning for Mamba models?
>
> Sure, we have added methods from the MambaPEFT paper to the ICL results (Figure 9).
>
> > 6. A tangential comment...
>
> Thank you for the reference.
>
> > authors overstate somewhat the implications of their work for reducing emissions.
>
> We have adjusted this.
>
> > Claims And Evidence: No
>
> We have proven deviations in the input states of MambaBlocks produce output states with bounded deviations, and demonstrated these empirically.  Can the reviewer please explain why our claims and evidence are not supported?
>
> # References
> [1] Gu, Albert, and Tri Dao. "Mamba: Linear-time sequence modeling with selective state spaces." arXiv preprint arXiv:2312.00752 (2023).
>
> [2] Mikhaeil, J., Monfared, Z., and Durstewitz, D. On the difficulty of learning chaotic dynamics with rnns. Advances in Neural Information Processing Systems, 35: 11297–11312, 2022
>
> [3] N.V. Kuznetsov, G.A. Leonov, On stability by the first approximation for discrete systems, 2005 International Conference on Physics and Control, PhysCon 2005, Proceedings Vol. 2005 (IEEE, 2005), p. 596

---

> > ### Comment · Reviewer_qxVf · 2025-07-14
> >
> > Thank you for your response. I would like to clarify a few of my comments and questions.
> >
> > 1. Whether or not the constraint that $A_{ii} \leq 0$ is referred to as an assumption of Theorem 1 is a matter of semantics. My point is merely that, as this is a design choice that is required in the proof of Theorem 1, it should be explicitly mentioned either in the statement of the theorem or when the authors introduce the Mamba architecture. This is important to make clear what in the model architecture fundamentally leads to stability.
> >
> > 2. It is still not clear to me why a direct proof does not suffice. So long as $\tilde{B}_t$ is bounded from above by a constant, or another explicit function of time that grows no faster than $\exp(-T \zeta)$ for $\zeta$ as in my prior comment, one will still have a bound of the form I sketched, no? I emphasize that I did not intend to give a complete proof recapitulating the results you present, just to sketch why I think this approach gives equivalent bounds since the system is linear.
> >
> > 3. I thank the authors for adding error bars, but have several questions about the resulting figures. The standard deviations shown in Figure 2 are huge, and the error bars often include zero. Are there actually experiments that give negative divergence, or is this an artefact? If the distribution of divergences is skewed, showing the standard deviation is not very statistically informative, and I'd suggest showing asymmetric confidence intervals generated using bootstrapping. Also, can you show error bars in Figure 4?

---

> > > ### Author Response · Authors · 2025-07-18
> > > **Reply**
> > >
> > > Dear Reviewer qxVf,
> > >
> > > > Whether or not the constraint that $A_{ii} \leq 0$ is referred to as an assumption of Theorem 1 is a matter of semantics...
> > >
> > > We will incorporate this change.
> > >
> > > > It is still not clear to me why a direct proof does not suffice. So long as $\tilde{B}_t$ is bounded from above by a constant,
> > >
> > > Yes, but the reviewer is conflating different notions of stability.  I.e., assuming the framework of Lyapunov stability, we have:
> > >
> > > $\max | F_{\theta}^N( x_{0}, u_1) - F_{\theta}^N( x_{0} + \epsilon, u_1+ \epsilon)| \in \mathcal{O}{(\epsilon \exp{(N \lambda ))}}.$
> > >
> > > The divergence in epsilon close inputs is exactly what the paper is trying to characterize.  Furthermore, we earlier remarked the following:
> > > > If considering some alternate form of stability, the bounds in Theorem 1, 2, 3, and 4 would not be readily available.
> > > Finally, we ask the reviewer to please look at the proofs for Theorems 1, 2, 3, and 4. There is much more work to prove the various theoretical results after deriving Lemma 1.
> > >
> > > To be clear, the use of Lyapunov stability allows us to directly characterize the phenomena we are interested in, and its use is not a valid criticism.  Perhaps, with more work, the line of derivations the reviewer has provided (which we note borrows heavily from our nontrivial proof in Lemma 1) could be continued to potentially characterize some similar divergence behavior in the inputs of Mamba models.  But as it stands, Lyapunov stability directly supplies this without excessive additional work.
> > >
> > > > I thank the authors for adding error bars, but have several questions about the resulting figures. The standard deviations shown in Figure 2 are huge, and the error bars often include zero. Are there actually experiments that give negative divergence, or is this an artefact? If the distribution of divergences is skewed, showing the standard deviation is not very statistically informative, and I'd suggest showing asymmetric confidence intervals generated using bootstrapping. Also, can you show error bars in Figure 4?
> > >
> > > This is an artifact of how yerr/standard deviations are plotted in matplotlib, please see [here](https://matplotlib.org/stable/api/_as_gen/matplotlib.pyplot.errorbar.html) for more details.  As stated in the paper, mean divergence is nonnegative:
> > > > The difference in model performance is reported as the mean divergence (i.e., absolute difference) between the original full-precision and respective mixed-precision model,
> > >
> > > Huge is a relative term, e.g., full precision full-finetuning Winogrande zero-shot performance for Pythia 410M is 52.57.  But indeed, we agree, the point of the plots is to show that Transformer LLMs' performance can vary drastically based on MPFT+LoRA configurations, whereas, in contrast, Mamba models are significantly more robust to these fine-tuning frameworks (as shown by their significantly smaller error bars).  We will add error bars to Figure 1.

---

> > > > ### Comment · Reviewer_qxVf · 2025-07-22
> > > >
> > > > Thank you for your continued, patient responses.
> > > >
> > > > - I'm sorry that I did not clearly motivate my line of questioning, but the fundamental point of my suggestion is that it is possible to analyze the stability of Mamba in a similar direct way to how one would analyze a linear RNN with time-invariant weights. That is, the fact that the weights are time-variant does not mean that one needs to rely on general machinery, and to be frank I disagree with the authors that the proof is particularly complex. All of that said, I will not further pursue this point; the discussion can be left here.
> > > >
> > > > - Given that the deviations are all non-negative, it is not statistically informative to show the error bars as in the current version of the paper. I would ask that you revise Figure 2 to use error bars that respect the sign of the deviations. Also, is there any possibility of adding error bars to other results, e.g. Figure 4?

---

> > > > > ### Author Response · Authors · 2025-07-23
> > > > > **Figure updates**
> > > > >
> > > > > Thank you for the reply.
> > > > >
> > > > > > Given that the deviations are all non-negative, it is not statistically informative to show the error bars as in the current version of the paper. I would ask that you revise Figure 2 to use error bars that respect the sign of the deviations. Also, is there any possibility of adding error bars to other results, e.g. Figure 4?
> > > > >
> > > > > The suggested edits have been made to both figures and are available in the latest revision.

---

### Review · Reviewer_SpRc · 2025-07-05

**Summary Of Contributions:**

The goal of this paper is to understand the robustness of models based on SSMs such as Mamba, both theoretically and empirically. Theoretically, this paper identifies that prior works analyze SSMs via time-invariance property, but Mamba has time-varying parameters, so prior analysis is not enough. To address this issue, authors use theory from dynamic systems to prove the robustness of Mamba models, by showing the maximal Lyapunov exponent $\lambda_{\max}\leq 0$, this shows that the input perturbations won't cause exponentially compounded error over time. Next, they perform extensive experiments to show the robustness of fine-tuning for Mamba models compared to transformer-based models (Pythia and OpenELM). Experiments include: various modes of fine-tuning, such as full-precision, mixed-precision, all parameters, LoRA, SLL LoRA. Mamba models exhibit much constrained divergence behavior of different fine-tuning procedures compared to Pythia and OpenELM, particularly obvious on Winogrande. Further, authors show that Mamba models, while achieve lower 5-shot MMLU accuracy, are more stable under choices of hyperparameters. Finally, authors empirically verify that under instruction tuning, Mamba exhibits improved ICL ability compared to Pythia.

**Audience:**

Yes

**Broader Impact Concerns:**

N/A.

**Claims And Evidence:**

Yes

**Requested Changes:**

I think the paper could be significantly improved if it emphasizes more on the main topic -- stability of fine-tuning for Mamba models -- in the introduction and in particular the authors could try to tie the theory and experiments on mixed precision fine-tuning more tightly. At its current form, the paper feels a bit fragmented (e.g., the consequences of the theory results could be further emphasized).

Finally: I'm not an expert in Mamba fine-tuning experiments, I would weigh in the opinions of the reviewers that are more familiar with this aspect.

**Strengths And Weaknesses:**

Strengths:

1. State-space models such as Mamba are important due to their efficiency compared to their transformer counterpart. However, their overall performances are weaker than transformers. The strongest contribution of this paper is to showcase the robustness and stability of Mamba during fine-tuning phase over transformer models.

2. This paper uses dynamical system theory to provide fundamental reasons why Mamba models are robust under fine-tuning, as input perturbation won't cause the error to exponentially grow, as $\lambda_{\max}\leq 0$ and the error is in the form of $\epsilon\cdot \exp(N \lambda_{\max})$ where $N$ is the number of applications of the model $F_\theta$.

3. Experiments are comprehensive, in the sense that various modes of fine-tuning are tested, including all parameters, FP, BF, parameter-efficient ones (LoRA, SLL LoRA) and the metric is the mean per-shot FP-32 divergence. Grid search over the set of hyperparameters (learning rate, LoRA dimension, warm ups) is performed to show the stability of Mamba. Finally, multi-shots performance improvements over zero-shot ICL after instruction-tuning has been tested, to show the improved performance of Mamba models under tuning.

Weaknesses:

1. The theory is explanatory and mainly applies existing results from dynamical systems, therefore it lacks novelty. However, it should be noted that the main purpose of the paper is to explain stability behavior of Mamba, rather developing new theory for dynamical systems.

2. I found the writing of the paper, in particular the core message, not super clear before the conclusion section. What is the major question posed in this paper that authors try to solve? Is that the stability behavior of Mamba for fine-tuning? What is the main takeaway from authors study? Should we widely adapt Mamba models for fine-tuning and instruction-tuning? These questions become more obvious in the conclusion section, but are unclear before that. For example, when reading through the paper, I feel the contents are quite segmented and a bit incoherent. For example, the theory suggests that Mamba models are stable under input perturbations, then the experimental sections show that the performance of Mamba models are generally more stable in the sense that they have smaller deviations among the choices of hyperparameters and in comparison to FP fine-tuning when using LoRA-type fine-tuning? How do these two parts relate to each other beyond using different precisions? It feels like through experiments, Mamba models are more robust and stable beyond using different precision for fine-tuning, but the authors do not very explicitly spell out this part well.

---

> ### Author Response · Authors · 2025-07-14
> **Reply**
>
> Dear Reviewer SpRc,
>
> We thank you for your helpful feedback.  We have made several changes to address your comments, e.g.:
> - Overhauled the writing of the paper to more clearly articulate what the cohesive story is
> - Removed the LoRA weight-tying Theorem and experiment, which ultimately did not fit the overall narrative and contributed to the disjointedness of the original submission
> - Added additional theoretical results more tightly demonstrating how Mamba SSMs are stable under changes specifically introduced by PEFT/MPFT (Theorems 3 and 4), thus better tying presented theory with the original stability experiments
> - Added additional experiments directly perturbing the inputs to SSM layers and demonstrating the maximum deviations of the output states are exponentially decaying
> - Added an additional Background section to make the paper self-contained

---

> > ### Comment · Reviewer_SpRc · 2025-07-23
> >
> > I thank the authors for the update. I don't have further questions.

---

### Review · Reviewer_z2sE · 2025-07-06

**Summary Of Contributions:**

This paper systematically studies the property of the emerging selective space model (Mamb) from the perspective of the dynamical system theory.

Specifically, a theoretical analysis of recurrent stability via Lyapunov exponents is conducted. It shows that Mamba models exhibit non-positive Lyapunov exponents, making their recurrent dynamics stable under small perturbations.

Experiments on multiple datasets and settings show its effectiveness.

**Audience:**

Yes

**Claims And Evidence:**

Yes

**Requested Changes:**

Please refer to the weakness mentioned above.

Major concerns:

- While this paper systematically analyzes the stability of Mamba from a perspective of Lyapunov exponents, its generalization error upper bound is not analyzed, which negatively impact the soundness of this work.

- While Sec.2.1.1 discusses the consequences of automatic mixed-precision, it lacks a theoretical analysis.

- Besides, more experimental evidence on this aspect is also needed.

- While this paper presents good experimental analysis by numerical comparison, it lacks a in-depth analysis from the feature space perspective.

- Can this paper discuss some possible application or user cases?

- The related work section need to be reorganized. For each aspect and research line, it should cover an independent subsection.

- This paper lacks a failure case and limitation discussion.

- The ICL gains of Mamba after instruction tuning are strong, but the pre-tuning ICL results are weak. This raises the question of architectural adequacy: is the ICL capability truly a learned emergent behavior or a by-product of alignment?

- It’s unclear how much the performance of instruction-tuned Mamba-2 (surpassing Pythia) is due to stability versus other factors like parameter count or tuning schedule.

Minor comments for improvement:

- The abstract should not contain citations.

- In the introduction, the authors directly start to mention Mambaa-2 models. But a clear definition is missing.

- Fig.2 should be on top of the page.

**Strengths And Weaknesses:**

Strength:

+ Overall this paper is easy-to-follow and clearly presented.

+ The use of Lyapunov exponents to establish formal stability is a novel and solid perspective.

+ The empirical studies are extensive, covering multiple model sizes, datasets, and fine-tuning schemes.

Weakness:

Major concerns:

- While this paper systematically analyzes the stability of Mamba from a perspective of Lyapunov exponents, its generalization error upper bound is not analyzed, which negatively impact the soundness of this work.

- While Sec.2.1.1 discusses the consequences of automatic mixed-precision, it lacks a theoretical analysis.

- Besides, more experimental evidence on this aspect is also needed.

- While this paper presents good experimental analysis by numerical comparison, it lacks a in-depth analysis from the feature space perspective.

- Can this paper discuss some possible application or user cases?

- The related work section need to be reorganized. For each aspect and research line, it should cover an independent subsection.

- This paper lacks a failure case and limitation discussion.

- The ICL gains of Mamba after instruction tuning are strong, but the pre-tuning ICL results are weak. This raises the question of architectural adequacy: is the ICL capability truly a learned emergent behavior or a by-product of alignment?

- It’s unclear how much the performance of instruction-tuned Mamba-2 (surpassing Pythia) is due to stability versus other factors like parameter count or tuning schedule.

Minor comments for improvement:

- The abstract should not contain citations.

- In the introduction, the authors directly start to mention Mambaa-2 models. But a clear definition is missing.

- Fig.2 should be on top of the page.

---

> ### Author Response · Authors · 2025-07-14
> **Reply**
>
> Dear Reviewer z2sE06,
>
> We thank you for the extensive review, we have updated the submission to reflect changes incorporating your feedback.  In addressing other reviewer feedback, we have also refocused the paper on Lyapunov stability (removing the weight-tying results, which did not fit with the overall narrative).  Please find our direct replies below:
>
> > While Sec.2.1.1 discusses the consequences of automatic mixed-precision, it lacks a theoretical analysis.
>
> We thank the reviewer for pointing this out.  We have included new theoretical results (Theorems 3 and 4) explicitly proving the stability of Mamba SSMs under automatic mixed-precision and PEFT (in particular, LoRA), which now better applies to the original empirical experiments.
>
> > Besides, more experimental evidence on this aspect is also needed.  While this paper presents good experimental analysis by numerical comparison, it lacks a in-depth analysis from the feature space perspective.
>
> We thank the reviewer for pointing this out.  We have included additional experiments directly perturbing the inputs within the SSM layer (and demonstrating exponentially decreasing deviations in the outputs) in Section 4.1 and Appendix D.
>
> > The related work section need to be reorganized. For each aspect and research line, it should cover an independent subsection.
>
> We thank the reviewer for catching this, we've reorganized this section accordingly.
>
> > Can this paper discuss some possible application or user cases?
>
> We've added this discussion to the Section 6, e.g.: "Furthermore, our theoretical contributions open the door for several follow up studies. E.g., additional results building off Theorems 3 and 4 may tackle generalization error bounds and privacy fine-tuning (by
> considering a language modeling framework and calculating upper bounds under distributions of ε), and
> Mamba-specific decoding schemes (where new algorithms may exploit the derived deviation bounds ensure
> next-token predictions lie within a deviation tolerance from the true optimal decoding)."
>
> > It’s unclear how much the performance of instruction-tuned Mamba-2 (surpassing Pythia) is due to stability versus other factors like parameter count or tuning schedule.  The ICL gains of Mamba after instruction tuning are strong, but the pre-tuning ICL results are weak. This raises the question of architectural adequacy: is the ICL capability truly a learned emergent behavior or a by-product of alignment?
>
> These are good points.  However, we note that the paper shows Mamba models are also stable to hyperparameter tuning (Section 4.3) and that the utilized ICL instruction-tuning hyperparameters were optimized for Transformer LLMs.  Thus, if there was a fundamental architectural bias in the tuning schedule, it should be in the favor of Pythia.
>
> We thank the reviewer for raising this question.  We note that recent studies [1] have claimed Mamba SSMs cannot perform ICL (and used synthetic data to demonstrate this claim, although Mamba-2 SSMs were subsequently demonstrated to perfectly solve such synthetic tasks [2]).  Answering this question is ultimately outside the scope of the paper.  We base our definition of emergent ability on [3], which makes the case that abilities displayed at certain parameter counts acquired during fine-tuning are still emergent.
>
> > While this paper systematically analyzes the stability of Mamba from a perspective of Lyapunov exponents, its generalization error upper bound is not analyzed, which negatively impact the soundness of this work.
>
> We thank the reviewer for raising this point.  We agree, leveraging the derived maximum deviation upper bounds for generalization error (as opposed to stability) would be an interesting contribution.  However, such work can/should be a paper in and of itself, which we allude to in future work.
>
> # References
> [1] Jongho Park, Jaeseung Park, Zheyang Xiong, Nayoung Lee, Jaewoong Cho, Samet Oymak, Kangwook Lee,
> and Dimitris Papailiopoulos. Can mamba learn how to learn? a comparative study on in-context learning
> tasks. International Conference on Machine Learning (ICML), 2024.
>
> [2] Dao, Tri, and Albert Gu. "Transformers are ssms: Generalized models and efficient algorithms through structured state space duality." arXiv preprint arXiv:2405.21060 (2024).
>
> [3] Wei, Jason, et al. "Emergent abilities of large language models." arXiv preprint arXiv:2206.07682 (2022).

---

> > ### Comment · Reviewer_z2sE · 2025-07-22
> > **Re: Reply**
> >
> > Dear Authors,
> >
> > Thanks for the response letter.
> >
> > The critical concerns have been addressed.
> >
> > I have no further questions.

---

### Decision · Action_Editor_pHfX · 2025-08-18

**Recommendation:** Accept as is

**Additional Comments:**

Across the reviews and discussions, there is clear appreciation for the core premise of the paper—namely, its effort to understand the stability of SSM Mamba models. Reviewers are aligned on the relevance and importance of this direction. However, many concerns were raised regarding presentation quality, experimental rigor, and on figures. The authors have addressed most of these points in their revisions.

I would recommend acceptance of the paper in its current form. That said, in the final version, please ensure any remaining loose ends are resolved (Figures 7 and 8 in Appendix for example).

**Audience:**

Yes

**Audience Explanation:**

The paper’s themes are closely aligned with several active areas of interest in the LLM landscape.

**Claims And Evidence:**

Yes

**Claims Explanation:**

The paper investigates stability concerns in state space models, focusing specifically on recurrent stability characterized through Lyapunov exponents. Theorems 2, 3, and 4 rigorously establish the stability properties, providing theoretical grounding for the authors' claims. These findings are further reinforced by empirical results across diverse datasets.